# Resolving seasonal and diel dynamics of non-rainfall water inputs in a Mediterranean ecosystem using lysimeters

Sinikka Jasmin Paulus[1, 3], Tarek Sebastian El-Madany[1], René Orth[1], Anke Hildebrandt[2, 3], Thomas Wutzler[1], Arnaud Carrara[4], Gerardo Moreno[5], Oscar Perez-Priego[6], Olaf Kolle[1], Markus Reichstein[1], and Mirco Migliavacca[1, 7]

[1]Max Planck Institute for Biogeochemistry, Department of Biogeochemical Integration, Jena, Germany
[2]Helmholtz Centre for Environmental Research (UFZ), Department Computational Hydrosystems, Leipzig, Germany
[3]Friedrich Schiller University, Institute of Geoscience, Jena, Germany
[4]Fundacion Centro de Estudios Ambientales del Mediterráneo (CEAM), Valencia, Spain
[5]Universidad de Extremadura, Institute for Silvopastoralism Research (INDEHESA), Plasencia , Spain
[6]Universidad de Córdoba, Department of Forestry Engineering, Córdoba, Spain
[7]current address: European Commission Joint Research Centre (JRC), Bio-Economy Unit, Ispra, Italy

**Correspondence:** Sinikka Paulus (spaulus@bgc-jena.mpg.de)

**Abstract.** The input of liquid water to terrestrial ecosystems is composed of rain and non-rainfall water (NRW). The latter comprises dew, fog, and adsorption of atmospheric vapor on soil particle surfaces. Although NRW inputs can be relevant to support ecosystem functioning in seasonally dry ecosystems, they are understudied, being relatively small, and therefore hard to measure. In this study, we apply a partitioning routine focusing on NRW inputs over one year of data from large high-precision

weighing lysimeters at a semi-arid Mediterranean site. NRW inputs occur for at least $3\,\mathrm{h}$ on $297\,\mathrm{days}$ ($81\,\%$ of the year) with a mean diel duration of $6\,\mathrm{hours}$. They reflect a pronounced seasonality as modulated by environmental conditions (i.e., temperature and net radiation). During the wet season, both dew and fog dominate NRW, while during the dry season it is soil adsorption of atmospheric water vapor. Although NRW contributes only $7.4\,\%$ to the annual water input NRW is the only water input to the ecosystem during $15\,\mathrm{weeks}$, mainly in the dry season. Benefitting from the comprehensive set of measurements

at our experimental site, we show that our findings are in line with (i) independent measurements, and (ii) independent model simulations forced with (near-) surface energy and moisture measurements. Further, we discuss the simultaneous occurrence of soil vapor adsorption and negative eddy covariance-derived latent heat fluxes. This study shows that NRW inputs can be reliably detected through high-resolution weighing lysimeters and a few additional measurements. Their main occurrence during night-time underlines the necessity to consider ecosystem water fluxes at high temporal resolution and with 24-hour

coverage.

## 1   Introduction

Water availability at the land surface controls a variety of processes related to land-atmosphere exchange, such as the warming of the surface and air temperature ($T_a$, $°C$) (Seneviratne et al., 2010; Panwar et al., 2019), ecosystem carbon fluxes (Reichstein et al., 2007; El-Madany et al., 2021), and evapotranspiration ($ET$, mm) (Jung et al., 2010; Rodriguez-Iturbe et al.,

2001). Therefore, precise quantification of the water balance is crucial for understanding and simulating these processes. The largest atmospheric input of liquid water to ecosystems globally is rain. In addition, other liquid water inputs exist, which are summarized under the term non-rainfall water (NRW).

NRW comprises several types of processes: deposition (fog), condensation (formation of dew, soil distillation, and hoar frost), and soil vapor adsorption (Kidron and Starinsky, 2019). These processes are mainly controlled by i) atmospheric vapor
pressure ($e_a$, $hPa$), ii) surface temperature ($T_s$, $°C$) and iii) topsoil water potential. Fog is defined as water droplets suspended in the air at a concentration that reduces visibility below $1000\,\mathrm{m}$ (Glickman and Zenk, 2000). The droplets adhere to surfaces after contact. Fog occurrence is commonly inferred from relative humidity ($rH$, %), visibility measurements (Feigenwinter et al., 2020; Zhang et al., 2019b), or by using net longwave radiation measurements (Wærsted et al., 2017). Condensation processes are induced by $T_s$ decreasing below the dewpoint temperature ($T_{dew}$, $°C$) of surrounding air. This is referred to
either as dew when the water originates from the atmosphere, or soil distillation when the water originates from the soil beneath (Monteith, 1957). However, most literature summarizes both processes as dew because a distinction requires additional measurements such as stable water isotopes (Li et al., 2021b). Dew is often measured with lysimeters (Meissner et al., 2000; Groh et al., 2018; Zhang et al., 2019b) or by leaf wetness sensors (Feigenwinter et al., 2020).

In close contact with soil, the water vapor in the soil air is influenced by capillary and adsorptive forces (Tuller et al., 1999).
These forces increase in relevance as the soil dries out. They reduce the saturation vapor pressure ($e_{sat}$, kPa) as a function of soil dryness (Edlefsen et al., 1943), leading to an earlier phase change of water from vapor to liquid within the soil, compared to free air. Under such conditions, the $rH$ value of the near-surface air outside the soil is well below $100\,\%$, and condensation processes are classically not considered, but soil air can condensate. The respective NRW is the adsorption of atmospheric vapor. However, despite being well understood at pore and laboratory scales (Tuller et al., 1999; Arthur et al., 2016), this
process was underrepresented in NRW studies (Zhang et al., 2019b; Saaltink et al., 2020; Kohfahl et al., 2019).

So far, no standard technique has been developed to measure the different components of NRW inputs in the field. Partly, this is related to technical limitations in measuring capabilities (summarized in Feigenwinter et al., 2020). Modeling NRW inputs also remains a challenge, although a distinction must be made between modeling the frequency and duration of occurrence, and modeling the yields of the different NRW fluxes. In the case of dew for example fewer studies have addressed the development
of the latter models due to the overall complexity of heat and radiation exchange and many unknown necessary parameters (Tomaszkiewicz et al., 2015). The same is true for adsorption (Verhoef et al., 2006).

Furthermore, the relatively small contribution of NRW for the annual water balance lead to an underestimation of their relevance for ecosystem functioning. But NRW subsidies can extend or sustain ecosystem functioning when rain and soil moisture are low (Weathers et al., 2020; Tomaszkiewicz et al., 2015). This is long known for arid ecosystems (Duvdevani,
1964; Agam and Berliner, 2006; Kidron and Starinsky, 2019) but an increasing number of studies suggest ecological relevance also in the context of seasonal dry spells (Jacobs et al., 2006; Li et al., 2021b; Groh et al., 2018; Dawson and Goldsmith, 2018; Riedl et al., 2022). As leaf wetting events, dew and fog can affect leaf water status directly and alter the surface water and energy balance by changing leaf temperature, albedo, and water vapor deficit (Dawson and Goldsmith, 2018; Gerlein-Safdi et al., 2018; Aparecido et al., 2017). Uptake of water at the leaf surface was observed across ecosystems, species, and

plant types relieving daily water and carbon stress (Dawson and Goldsmith, 2018; Berry et al., 2014; Aparecido et al., 2017). Secondary effects of NRW inputs related to changes in canopy micro-climate have also been shown to reduce plant water stress and increase water use efficiency (Ben-Asher et al., 2010). Respiration in dry seasons of organisms like biocrusts (Kidron and Kronenfeld, 2020a), microbes within the soil (McHugh et al., 2015), and on standing litter (Evans et al., 2019; Gliksman et al., 2017) was sustained due to dew and adsorption. Nevertheless, the role of NRW inputs in connection with the carbon cycle across ecosystems has not yet been fully understood and quantified (Dawson and Goldsmith, 2018; Weathers et al., 2020; Kidron and Starinsky, 2019).

Research about the role of NRW remains limited by too few measurements of their actual amounts (Berry et al., 2019). Appropriate measurement facilities to analyze and quantify the local circumstances of NRW occurrence seem sparse. Past studies were often based on micro-lysimeters covering time periods of few weeks to some months. It was recently suggested that the temperature regimes in many formerly used micro-lysimeter setups deviated from the surrounding soil causing an over-estimation of the measured NRW (Kidron and Kronenfeld, 2020b). These issues are partly overcome for large high precision weighing lysimeters, where lower boundary control systems were developed which equilibrate soil temperature and moisture content between the inside of the lysimeter and the surrounding soil to prevent biases (Groh et al., 2016). Therefore, data collected with large weighing lysimeters can further contribute to the identification and quantification of NRW inputs. Yet, relatively few stations are located in semi-arid and arid environments (Perez-Priego et al., 2017; Dijkema et al., 2018; Kohfahl et al., 2019; Zhang et al., 2019b) where NRW inputs are expected to be particularly relevant.

In this study, we implement and test a processing scheme for identifying and quantifying NRW inputs in a seasonally dry ecosystem in continental Spain. The aims of this paper are to (i) analyze the seasonal NRW dynamics and their contribution to the annual water input at the site; and (ii) evaluate our results against independent observations and empirical models.

## 2 Material and Methods

### 2.1 Study site

All data investigated in this study originate from the experimental field site Majadas de Tietar, Caceres in Extremadura, Spain ($39°56'25.12''$ N, $5°46'28.70''$ W). All variables used are summarized in Table B. Average diel $T_a$ is $16.7\,°C$ with diel minimum and maximum $T_a$ of $3.1\,°C$ to $12.5\,°C$ in January and $18.6\,°C$ to $39.8\,°C$ in August. The rain mainly falls between October and April, with mean annual amounts of ca. $650\,mm$, with large interannual variations (El-Madany et al., 2020). The mean annual potential evapotranspiration calculated with the Priestley-Taylor equation (Knauer et al., 2018) amounts to $1524\,mm\,year^{-1}$. The ecosystem is a typical Mediterranean semi-arid tree-grass ecosystem (*Dehesa*) with low-density oak tree cover (*Quercus ilex* (L.), $\sim 20$ trees $ha^{-1}$) (Bogdanovich et al., 2021).

The herbaceous layer consists of native annual grasses, forbs, and legumes (Migliavacca et al., 2017). The growing season for the herbaceous layer begins after the first rains after summer (typically in mid-October) and is inhibited by low temperatures in winter before peaking in spring before the dry season (Luo et al., 2020). During the dry season, the herbaceous species are inactive until the return of rain (Perez-Priego et al., 2018), and bare soil is visible below the dry biomass. The mean leaf area

index of the herbaceous vegetation ranges between $0.25 \pm 0.07 \, \mathrm{m^2 \, m^{-2}}$ in summer to $1.75 \pm 0.25 \, \mathrm{m^2 \, m^{-2}}$ at the peak of the growing season in spring (El-Madany et al., 2021) with the same seasonal dynamics in vegetation height that varies between 90 $0.05 \, \mathrm{m}$ to $0.20 \, \mathrm{m}$ (Migliavacca et al., 2017). Biocrusts are not present at the site.

(Perez-Priego et al., 2018). The site is managed with low-intensity grazing by cows during the growing season (El-Madany et al., 2018). An exclusion fence of $\sim 50 \, \mathrm{cm}$ height was used to avoid cows stepping into the lysimeters. However, the fence was close enough to allow grazing of the columns in order to maintain the vegetation on lysimeters comparable with the rest of the plot (picture shown in Fig. A1). The soil is formed of alluvial deposits and classified as Abruptic Luvisol (IUSS Working 95 Group WRB, 2014) with sandy topsoil of $74\,\%$ sand, $20\,\%$ silt, and $6\,\%$ clay (Nair et al., 2019). A clay layer rests at a variable depth between $30 \, \mathrm{cm}$ to $100 \, \mathrm{cm}$. Although the trees also play a role in the water balance at the ecosystem scale, herbaceous vegetation dominates $ET$ (Perez-Priego et al., 2017). This work focuses on the water fluxes in open areas where lysimeters are located (Migliavacca et al., 2017).

### 2.1.1 Lysimeter technical specifications

The site is equipped with three lysimeter stations, each containing two weighable high-precision polyethylen-high-density lysimeters (UGT, Müncheberg, Germany), for a total of 6 columns. Each column has a $1 \, \mathrm{m^2}$ surface area and $1.20 \, \mathrm{m}$ column depth and is situated on a weighing system consisting of three precision shear-stress cells, respectively (Model 3510, Stainless Steel Shear Beam Load Cell, VPG Transducers, Heilbronn, Germany). The weight measurements are collected every $1 \, \mathrm{min}$ with a measurement precision of $10 \, \mathrm{g}$ ($0.01 \, \mathrm{mm}$). Each lysimeter station is equipped with a lower boundary control system 105 to avoid deviations from natural conditions due to the isolation of the lysimeter columns (Groh et al., 2016). Porous ceramic bars at the bottom of the lysimeters maintain the soil water potential within the column comparable with the soil surrounding them. Soil temperature ($T_{soil}$, °C) is controlled with a heat exchange system. Buried at the bottom of each lysimeter, $6 \, \mathrm{m}$ tubing systems are connected to tubing systems buried at the same depth in the surrounding soil. Pumping of water through the system in a closed loop regulates the soil temperature within each lysimeter column by heat exchange (Podlasly and Schwärzel, 2013). 110 $SWC$ and $T_{soil}$ are measured every $15 \, \mathrm{min}$ within the columns at 0.1, 0.3, 0.75, and $1 \, \mathrm{m}$ depth (UMP-1, Umwelt-Geräte-Technik GmbH, Müncheberg, Germany). Soil matric potential ($\Psi$, hPa) is measured every $15 \, \mathrm{min}$ at $0.1 \, \mathrm{m}$ depth with a porous ceramic cone full range pF meter (ecoTech Umwelt-Meßsysteme GmbH, Bonn, Germany).

The lysimeters were installed in 2015 by excavating undisturbed soil monoliths from open grassland areas. General aspects of the excavation method are described in Reth et al. (2021). The natural herbaceous vegetation cover was preserved. Pictures 115 of the lysimeter columns and an aerial photograph of the site and experimental set-up are shown in Fig. A1. The stations have 104 m, 91 m, and $24 \, \mathrm{m}$ distance to each other, respectively. The closest tree is $\sim 9 \, \mathrm{m}$ away. We analyze a period of one year from June 1st, 2019 to May 31st, 2020.

### 2.1.2 Ancillary measurements outside the lysimeters

For partitioning the lysimeter weight changes ($\Delta W$, $\mathrm{kg \, min^{-1}}$) into different water fluxes and modeling (see Section 2.2.1, 120 2.3.2, and 2.3.3) we used additional field measurements collected every $30 \, \mathrm{min}$. Meteorological variables monitored are rain,

which is measured with a weighing rain gauge (TRwS 514 precipitation sensor, MPS systém, Slovakia), $T_a$, and $rH$ at 1 m (Pt–100 and capacitive humidity sensor CPK1–5, MELA Sensortechnik, Germany). $T_{dew}$ was calculated based on $T_a$ and $rH$ (see Appendix C). Actual vapor pressure ($e_a$, hPa) is calculated from $T_a$ and $rH$.

Short- ($SW$, $\mathrm{W\,m^{-2}}$) and longwave ($LW$, $\mathrm{W\,m^{-2}}$) downwelling ($\downarrow$) and upwelling ($\uparrow$) radiation of the herbaceous layer was observed with a net radiometer (CNR4, Kipp and Zonen, Delft, The Netherlands) at a measurement height of $\sim 2\,\mathrm{m}$. $T_s$ is calculated from $LW$ (equations in Appendix C). Ground heat flux ($G$, $\mathrm{W\,m^{-2}}$) was monitored with soil heat flux plates (Heatflux Ramco HP3, McVan Instruments, Mulgrave, Australia).

$T_{soil}$, and $SWC$ were measured along a profile outside the lysimeters at 0.05 m, 0.10 m, and $0.2\,\mathrm{m}$ depth, respectively (Delta-ML3, Delta-T Devices Ltd, Burwell Cambridge, UK).

Fluxes of latent heat ($\lambda E$, $\mathrm{W\,m^{-2}}$), wind speed ($u$, $\mathrm{m\,s^{-1}}$) and friction velocity ($u_*$, $\mathrm{m\,s^{-1}}$) were measured by an eddy covariance (EC) system, consisting of a sonic anemometer (R3–50, Gill Instruments, Lymingon UK) and an infra-red gas analyzer (LI-7200, Licor Biosciences, Lincoln, USA) at $1.6\,\mathrm{m}$ sampling height and targeting the herbaceous layer (Perez-Priego et al., 2017). For further details on the EC data processing, see El-Madany et al. (2018).

## 2.2  Data analysis

### 2.2.1  Lysimeter data processing

The processing of lysimeter data comprises several steps: (a) raw weight data filtering, (b) time-series smoothing, and (c) flux partitioning. The processing workflow is displayed in Fig. 1. All code used in the analysis is available for reproducibility in the open-source R environment for statistical programming (R Core Team, 2020). See the data and code availability statement for more details.

**a) Raw data processing:** changes in the water reservoir through the lower boundary system and lysimeter column weights are added. Outliers are filtered out by setting plausible threshold values; $-0.5\,\mathrm{kg\,min^{-1}} < \Delta W < 1\,\mathrm{kg\,min^{-1}}$ (Schrader et al., 2013). Additionally, outliers within these threshold values were identified by comparing $\Delta W$ across the six columns. If $\Delta W$ is due to rain, we expect similar responses across lysimeters. In contrast, if only one lysimeter column shows an anomalous $\Delta W$ we considered this as an artifact (e.g., small animals such as snakes or rabbits stepping on the column or issues with the boundary control) that can be removed from the time series (Hannes et al., 2015). For identifying these values, we calculated the mean $\Delta W$ of all six lysimeters for an interval $i$ of one minute. This value was then subtracted from the individual $\Delta W$ measurements during $i$. The resulting value is a normalized weight change ($\Delta \mathrm{W}_{normalized,i}$, kg) for each column. Then, an average standard deviation ($\overline{\sigma}$, kg) was calculated from $\Delta \mathrm{W}_{normalized,i-3}$ to $\Delta \mathrm{W}_{normalized,i+3}$. $\Delta \mathrm{W}_{normalized,i} > (1.5 \cdot |\overline{\sigma}|)$ are replaced by not available (NA).

**b) Time-series smoothing:** is necessary to remove noise from the time series before the partitioning and data analysis based on $\Delta W$ (Schrader et al., 2013). Since noise in this type of data is not constant in time due to wind for example (Nolz et al., 2013), we apply a routine with adaptive averaging window widths ($\omega$) and adaptive $\Delta W$ thresholds ($\delta$) (AWAT) from Peters et al. (2014, 2016, 2017). As an intermediate result, the AWAT algorithm produces a step function of lysimeter weight. At

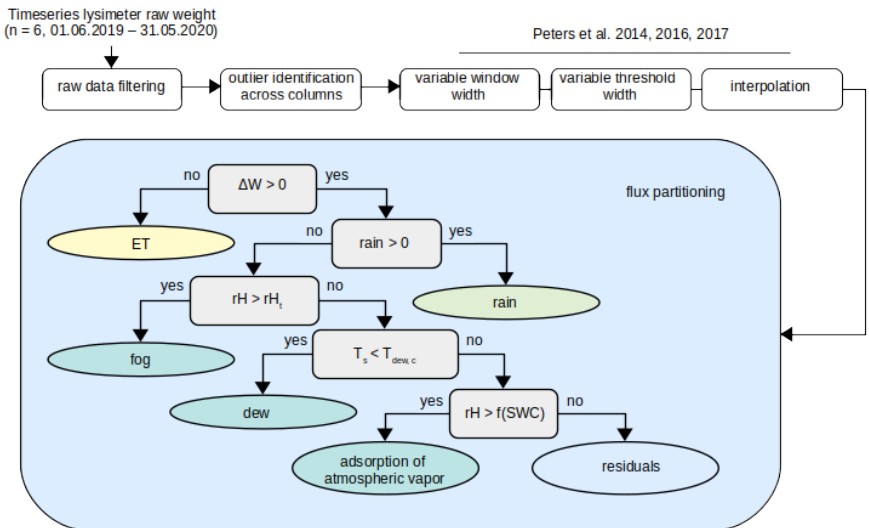

**Figure 1.** Overview of the lysimeter data processing to determine non-rainfall water inputs. White rectangles represent the steps of the pre-processing chain before flux partitioning. The pre-processed time series is then classified into the different vertical water fluxes based on a decision tree structure shown in the light blue box; grey rectangles stand for the decision nodes, and ellipses for the endpoint nodes. The decision tree is adapted from Zhang et al. (2019a). Detailed information on the choice of parameters $rH_t$, $T_{dewc}$ and f($SWC$), and their uncertainty are given in Chapter 2.2.1 and Chapter 4.

last, a smoothing of the $1\,\mathrm{min}$ resolution time series is performed using a spline interpolation. We chose this routine because
the authors included a processing step developed specifically for dew conditions (Peters et al., 2017). In our application, the parameter $\omega$ varied between $3\,\mathrm{min}$ to $31\,\mathrm{min}$ and $\delta$ between $0.01\,\mathrm{mm}$ to $0.05\,\mathrm{mm}$. A detailed overview of the algorithm and an evaluation of the performance are compiled in Peters et al. (2017) and Hannes et al. (2015). If one out of all columns was measuring more than $16\,\mathrm{h}$ of weight gain during one day, the full day was excluded for the analysis for the respective column. Due to technical problems that became obvious after data processing, lysimeter column number 4 was completely excluded
from further analysis. After the smoothing, the time series are aggregated to a 5-minute resolution to further decrease the remaining influence of noise for the subsequent step of flux partitioning, particularly for values close to zero.

     **c) Flux partitioning:** the filtered time series of $\Delta W$ is divided into six different water fluxes, assuming that in 5-minute intervals, only one process prevails over the others. Negative $\Delta W$ is always classified as $ET$. Positive $\Delta W$ is separated into different flux categories in a decision tree structure considering additional meteorological data, as illustrated in Fig. 1. At the
second decision node, we check if the rain gauge identifies a rain event during the period of weight increase. If true, all positive $\Delta W$ are classified as rain during $30\,\mathrm{min}$ before and after the event. This period was selected to match the measurement interval of rain gauges which is $30\,\mathrm{minutes}$. If false, e.g., in the absence of rain, $rH$ is evaluated.

     Theoretically, fog occurs at a $rH$ of $100\,\%$. We noticed, however, that the maximum saturation values varied depending on the sensor, with values between $98\,\%$ to $104\,\%$ (Fig. G1). We therefore decided to set a $rH$ threshold ($rH_t$) that is based on the

170 data distribution of the sensor to account for the individual uncertainty when the air is nearly saturated, for systematic biases, and for drifts. In our study $\Delta W$ is attributed to fog when $rH_t = 97.1\%$ which is the $90^{th}$ percentile of the $rH$ sensor records measured at $1\,\mathrm{m}$ height.

If neither fog nor rain is detected, we compared $T_s$ with $T_{dew}$. Between the top of the canopy and the soil surface, however, there is a large temperature gradient which is also reflected in dew amounts (Kidron, 2010). Since the installation height of the

175 sensor is $1\,\mathrm{m}$ we needed to approximate a value that better reflects $T_{dew}$ at the height of the condensation surfaces. Despite a reference height of $1\,\mathrm{cm}$ is generally used (Monteith, 1957), we approximated $10\,\mathrm{cm}$ which reflects the average canopy height of the herbaceous layer. To include the effect of the height difference, we compared sensors installed at $0.1\,\mathrm{m}$ and at $1\,\mathrm{m}$ height during a measurement campaign of two months in Spring 2021. The results show that the median temperature difference between the $0.1\,\mathrm{m}$ to $1\,\mathrm{m}$ sensor height is $1.4\,^{\circ}\mathrm{C}$ with an IQR of $1.2\,^{\circ}\mathrm{C}$. Dew was therefore assigned when $T_s < (T_{dew} -$

$T_{dew,c})$ where $T_{dew,c}$ is set to $1.4\,^{\circ}\mathrm{C}$. Note that the measurement height and the offset value should be site-specifically adapted depending on the height of the condensation surfaces.

If no dew was detected, $\Delta W$ could be potentially attributed to soil adsorption of atmospheric vapor (Zhang et al., 2019a). Adsorption occurs, however, at specific soil hydraulic and meteorological conditions given by a relation between $\Psi$ and atmospheric $rH$. Those have been observed in the laboratory (Camuffo, 1984; Arthur et al., 2016) and suggested from field

observations (Kosmas et al., 2001; Uclés et al., 2015; Zhang et al., 2019b). For adsorption $\Psi$ falls below (or is less negative than) a given threshold. This threshold can also be expressed in terms of $SWC$, given the relationship between $\Psi$ and $SWC$ through the soil water retention curve. In order to integrate this knowledge into the classification of adsorption, measurements of $rH$ and $SWC$ were analyzed for the periods where adsorption conditions were identified based on modeling (see section 2.3.3). For these periods, we fit a nonlinear quantile regression ($90^{th}$ *percentile*) to measurements of $rH$ and $SWC$. This

nonlinear relationship is depicted in Appendix Fig. H1. Although for the description of this relationship multiple empirical functions exist for samples measured in the laboratory under equilibrated conditions, they do not apply to our case due to the different observation ranges of $rH$ and $SWC$ at our site (see Fig. H1 for examples). When $rH \geq \mathrm{f}(SWC)$, soil adsorption of atmospheric vapor is assigned to positive $\Delta W$. When $rH < f(SWC)$ we classify the flux as residual. Example lysimeter weight evolution and the assigned flux categories for a week with and without rain is shown in Figure D1 in the Appendix.

Fluxes are presented in the results section as mean and standard deviation across the five lysimeter columns.

The results of the partitioning algorithm, as well as the uncertainty of the NRW inputs might be sensitive to the set of parameters and thresholds values used at each node. Therefore, the impact of the choice of threshold parameters on the flux uncertainty was tested. To do so, we defined a parameter set that describes the upper and lower limit of the parameter values, specifically $rH_t = [80, 100]\%$ and $T_{dew,c} = [0, 1.5]\,^{\circ}\mathrm{C}$. The parameter values tested cover the ranges of threshold values for

$rH$ and $T_s$ which are currently used in similar studies to identify fog and dew conditions (e.g., Feigenwinter et al., 2020; Zhang et al., 2019b). We then ran forward the partitioning algorithm with all the combinations of the parameters. The uncertainty of the calculated fluxes was then characterized based on the interquartile range (IQR) of the resulting flux quantities as will be shown in Section 3.1.

## 2.3 Quantitative validation and benchmarking of the NRW assignment

We test the plausibility of inferred water fluxes by comparison against direct measurements (for rain and $ET$), and in absence of respective direct measurements derived with alternative measured variables for NRW by benchmarking (for fog) and comparison against model predictions (for dew and adsorption).

### 2.3.1 Identification of fog conditions and their separation from dew

Since clear sky conditions and the amount of humidity present in the air is a key factor for both dew and radiation fog formation, their distinction is challenging. Measurements from visibility sensors are therefore used in addition to $rH_t$ to classify fog unambiguously (e.g. Feigenwinter et al., 2020; Riedl et al., 2022). Given that at the experimental site visibility sensors are not available, we use $LW$ to benchmark fog occurrence in our analysis.

When a fog layer contains a sufficient amount of liquid water the suspended droplets cause an optically thick layer obstructing $LW$ radiation from and to the sky (Wærsted et al., 2017). Under such conditions measurements of $LW\downarrow$ and $LW\uparrow$ within the fog layer should be identical. Therefore, we compare the ratio of $LW\downarrow$ and $LW\uparrow$ in periods classified as fog to the periods classified as dew from the $\Delta W$ partitioning. We used the non-parametric Mann-Whitney U test to evaluate statistically significant differences between the radiation patterns associated with both conditions. This was applied to a period of three years from January 1st, 2018 to December 31st, 2020 to increase the number of events for better statistical robustness.

### 2.3.2 Modelling dew

Dew is modeled as negative latent heat flux calculated based on models originally developed for determining evapotranspiration: (i) the Penman-Monteith (PM) (Monteith, 1965) equation which combines processes related to radiative energy and vapor pressure deficit (previously applied for dew in various forms by e.g. Jacobs et al., 2006; Aguirre-Gutiérrez et al., 2019; Groh et al., 2018), and (ii) equilibrium evaporation (previously applied for dew by Uclés et al., 2014).

We implemented the models as described in Ritter et al. (2019) and Jacobs et al. (2006) (Eq. (2.1)) and in Uclés et al. (2014) (Eq. (2.2)).

$$\lambda E = \frac{s}{s+\gamma} \cdot (R_{net} - G) + \frac{\gamma}{s+\gamma} \frac{\rho_a \, \gamma \, \Delta q}{r_{av}} \tag{2.1}$$

$$\lambda E = \frac{s \, (R_{net} - G)}{s+\gamma} \tag{2.2}$$

where $\lambda E$ ($\mathrm{W\,m^{-2}}$) is the latent heat flux, $s$ ($\mathrm{Pa\,K^{-1}}$) is the derivative of the saturation vapor pressure curve defined as ($d e_{sat}/dT$), $\gamma$ ($\mathrm{kPa\,K^{-1}}$) is the psychrometric constant, $R_{net}$ is net radiation ($\mathrm{W\,m^{-2}}$), $G$ is the soil heat flux ($\mathrm{W\,m^{-2}}$), $\rho_a$ ($\mathrm{kg\,m^{-3}}$) is the density of air, and $\Delta q$ is the deficit of specific humidity at reference level ($\mathrm{kg\,kg^{-1}}$). $r_{av}$ is the aerodynamic resistance to vapor transport between the surface and the air ($\mathrm{s\,m^{-1}}$) and was derived with an empirical relationship based on $u_*$ (Thom, 1972).

For both equations, dew occurs when $\lambda E < 0$ and $T_s \leq (T_{dew} -1.4\,°\mathrm{C})$ (as explained in section 2.2.1). This approach has been reported to be suitable for detecting potential dew conditions and to analyze dew frequency and duration. But it is limited in reproducing dew yields (Ritter et al., 2019). Hence we focus on comparing condition lengths rather than yields since we aim to validate dew detection by the partitioning routine. Measured flux durations were compared to respective model estimates modeled flux using correlation, mean absolute error (MAE), and root mean squared error (RMSE) (full equations Appendix C).

### 2.3.3 Modelling adsorption

Adsorption conditions were identified based on the vertical humidity gradient near the surface. We implemented the re-arranged aerodynamic diffusion equation originally used by Milly (1984) and previously applied for modeling adsorption by Verhoef et al. (2006):

$$e_{s,0} = \frac{\gamma\, r_{av}\, \lambda E}{\rho_a\, C_p} + e_a \tag{2.3}$$

The target value $e_{s,0}$ (kPa) is vapor pressure of soil air at the surface. $e_a$ (kPa) is vapor pressure of the atmosphere and $C_p$ ($\mathrm{J\,kg^{-1}\,K^{-1}}$) is the specific heat capacity of dry air at constant pressure. The other parameters are the same as in Eq. (2.1). When $e_{s,0} < e_a$, gradient driven vapor flow is towards the soil surface. This is assumed to be indicative of the adsorption of vapor from the atmosphere. Different from the dew models, we used high-quality filtered measurements from EC for $\lambda E$ in Eq. (2.3). Again, we only compared the simulated daily duration of suitable conditions of atmospheric adsorption against the results obtained with the lysimeter weights partitioning and we constrained the comparison to periods when $T_s \leq (T_{dew} -1.4\,°\mathrm{C})$.

## 3  Results

### 3.1  Diel and seasonal changes of water fluxes

Figure 2 shows the fingerprint of lysimeter $\Delta W$ induced by $ET$, rain, and NRW, assigned water flux types (exemplarily shown for lysimeter column 6) and $rH$, respectively, together with mean diel $SWC$ at $0.05\,\mathrm{m}$ depth and maximum diel $T_s$ range.

Lysimeter $\Delta W$ are mainly negative between sunrise and sunset (Fig. 2a), e.g., water is lost from the column due to $ET$. After sunset, however, they are zero or positive during most of the year, indicating water input. This diel pattern is consistent across seasons, following the seasonal daylight variability. The flux classification reveals seasonal differences in the prevailing NRW fluxes (Fig. 2b). At the beginning of June, atmospheric adsorption mainly occurs during the early morning, before sunrise. From July to September the length of the adsorption period increases, and the onset shifts towards earlier in the night. In this period, the diel variability of $rH$ is relatively low (Fig. 2c), $SWC$ at $5\,\mathrm{cm}$ depth is below $10\,\%$ and $T_s$ oscillates up to an amplitude of $35\,°\mathrm{C\,day^{-1}}$ (Fig. 2d & e). A rain event, in late July increases $SWC$ and is followed by some days of increased $ET$ which also prevails during night-time.

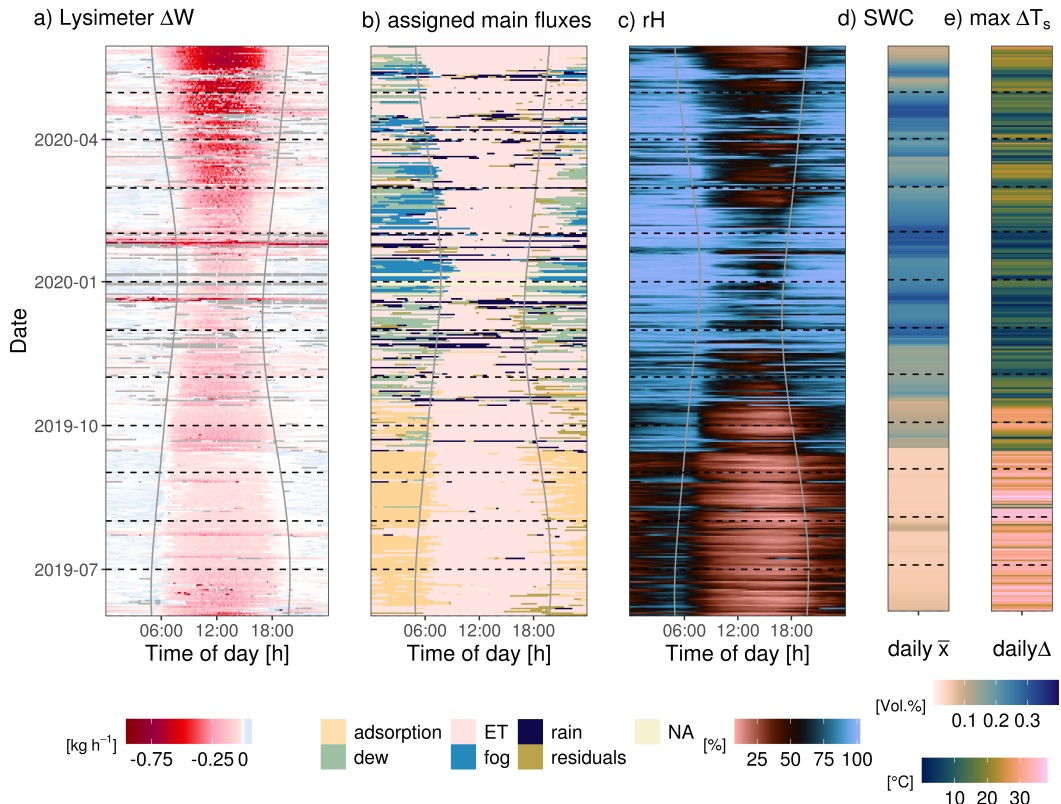

**Figure 2.** Diel and seasonal dynamics of a) lysimeter weight changes, b) assigned flux types, both from lysimeter 6, c) $rH$ measured at $1\,\mathrm{m}$ height. Solid vertical lines mark sunrise and sunset, respectively, determined with the geographic coordinates of the field site. Mean $SWC$ at $0.05\,\mathrm{m}$ depth and maximum diel difference of $T_s$ are displayed as diel measurements in panels d) and e).

A rain event in late September leads to longer-lasting increases in $SWC$ and $rH$. Such conditions are typically associated with vegetation re-greening. $ET$ and dew alternate during night-time. Frequent rain events in November and December are accompanied by dew and fog becoming the dominant NRW inputs until the end of the measurement period in April.

On most days, the average NRWI is $< 0.2\,\mathrm{mm\,d^{-1}}$ (Fig. E1). Yields of $< 0.2\,\mathrm{mm\,d^{-1}}$ are observed on 54 days distributed throughout the year. During the dry season NRWI is predominantly between 0.1 and $0.2\,\mathrm{mm\,d^{-1}}$ while over the wet season, the yields are typically small and $< 0.2\,\mathrm{mm\,d^{-1}}$. The maximum recorded daily NRWI was $2.42\,\mathrm{mm\,d^{-1}}$ on May 13th, 2020 between several small, consecutive rain events.

    NRW fluxes occur for at least $3\,\mathrm{h}$ during $297\,\mathrm{days}$ ($81\,\%$ of the year) with a mean duration of 6 hours per day of occurrence.

The weekly sums in Fig. 3 illustrate that the seasonal dynamics are consistent across lysimeters. The ecosystem receives atmospheric water at any time of the year but with shifting relative relevance of the water flux types during the wet and the dry season. Rain is the dominant liquid water input (i.e. it contributes more than $50\,\%$ of the weekly water input) during $29\,\mathrm{weeks}$

since its total amount is usually much greater than NRW inputs, whereas NRW inputs are dominant in $24$ weeks. They are even the only water input during $15$ weeks with adsorption as exclusive water input in $10$ weeks of the year during the dry season.

During the dry season, the median contribution of adsorption is $0.9\,\mathrm{mm\,week^{-1}}$ whereas dew and fog contribute $< 0.5\,\mathrm{mm\,week^{-1}}$. With $ET$ amounting on average to $5.7\,\mathrm{mm\,week^{-1}}$, thus adsorption compensates on average for $19\,\%$ of the weekly water loss through $ET$ during summer, ranging between $8.0\,\%$ at the beginning of June to $42.5\,\%$ at the beginning of September. During the wet season, the median contribution of dew and fog is $0.43\,\mathrm{mm\,week^{-1}}$ and $0.38\,\mathrm{mm\,week^{-1}}$, respectively. Dew and fog occurrence is synchronized with rain with regard to the seasonal occurrence (in the wet season), and therefore their relative

contribution to the water input on a weekly scale is small (see Fig. 3d)."

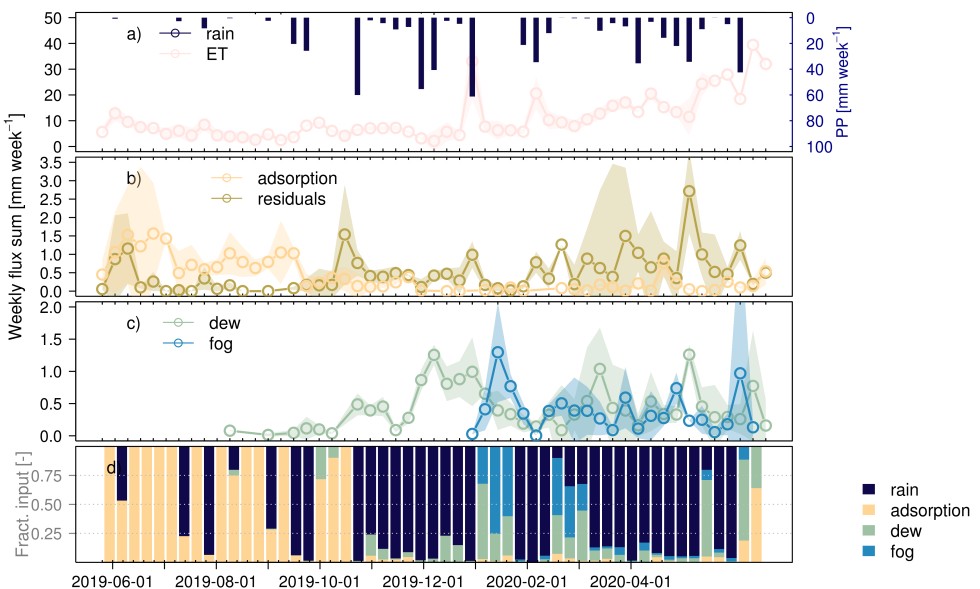

**Figure 3.** Median of weekly flux sums across lysimeters for (a) $ET$ and rain, (b) adsorption and noise, (c) dew and fog. (d) relative contributions of rain and NRW inputs to total water input per week, respectively. Shading indicates the variability across lysimeter columns expressed as an inter-quartile range (IQR).

## 3.2   Water fluxes and their consistency with measurements and theory

The cumulative measured rain is $565.0 \pm 11\,\mathrm{mm}$ as an average across the five lysimeters. Thereby, the difference between the maximum and minimum across the lysimeters is $30.7\,\mathrm{mm}$, which is an absolute deviation of $5\,\%$ between columns. For comparison, the rain gauge recorded $597\,\mathrm{mm}$ of rain during the same time period. The underestimation of the estimates from

lysimeters compared to the rain gauge, as well as the deviation between columns, is mainly caused by a few large rain events in January 2020 (Fig. 4a). The cumulative measured $ET$ across lysimeters is $570.7 \pm 20\,\mathrm{mm}$. The annual cumulated difference between lysimeters is $46.7\,\mathrm{mm}$. Annual cumulative $ET$ determined through EC measurements is $619.0\,\mathrm{mm}$. As in the case of

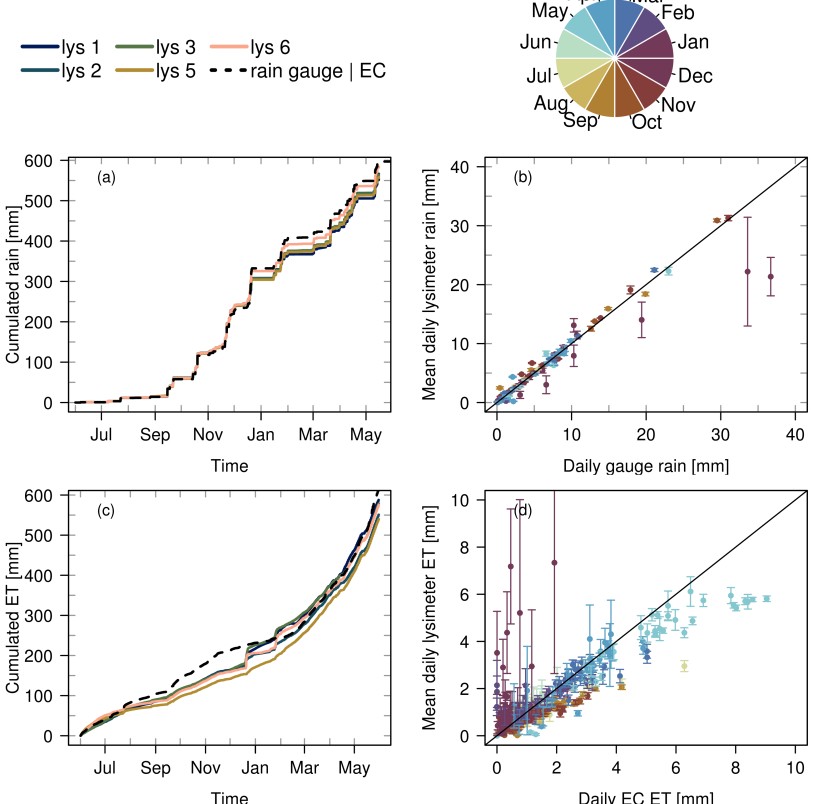

**Figure 4.** Comparison between cumulative (a,c) and mean diel (b,d) observations of rain and $ET$ between lysimeter and rain gauge (top row) and lysimeter and EC (bottom row). The points and vertical error bars indicate mean and standard deviation between lysimeter columns, which include measurement uncertainty and spatial variability.

rain, the $ET$ estimates deviate strongest in winter which indicates a technical problem during days with rain at this time of the year, while estimates are more consistent during the rest of the study period (Fig. 4d).

Descriptive statistics of the annual sums of NRW inputs across lysimeter columns are summarised in Table 1. The average annual NRW sum amounted to $41.9\,\mathrm{mm}$ with individual contributions of $6.6\,\mathrm{mm}$ from fog, $14.1\,\mathrm{mm}$ from dew, and $21.2\,\mathrm{mm}$ from adsorption. The differences in flux sums across lysimeters for all NRW inputs are relatively larger than for rain and $ET$, with coefficients of variation around $40\,\%$. The largest relative annual deviation between columns was found for adsorption with a difference of $20.0\,\mathrm{mm}$. But the flux which is most affected by the threshold parameter in the lysimeter flux partitioning is fog, which can be seen in Table 2. The IQR for the fluxes decreases the later in the partitioning scheme the respective flux is estimated (Fig. 1). Thereby, for dew and adsorption, the spatial variability between lysimeter columns (IQR in Table 1) exceeds the range of uncertainty related to the partitioning parameters (IQR in Table 2). The largest relative contribution to the mean total NRW input is adsorption with $50\,\%$, followed by dew with $34\,\%$ and fog with $16\,\%$.

**Table 1.** Mean, median, inter quartile range (IQR) and minimum, and maximum annual sum of individual NRW fluxes across lysimeter columns. All values are reported in $\mathrm{mm\,year^{-1}}$.

| Flux | Mean | Median | IQR | min | max |
|------|------|--------|-----|-----|-----|
| **Fog** | 6.6 | 7.6 | 3.9 | 3.1 | 9.5 |
| **Dew** | 14.1 | 11.8 | 8.6 | 9.5 | 20.2 |
| **Adsorption** | 21.2 | 16.6 | 8.3 | 14.4 | 34.5 |
| *Residuals* | *23.7* | *21.2* | *7.6* | *12.8* | *37.3* |

**Table 2.** Uncertainty related to the flux partitioning parameters. The table summarises the annual mean and median fluxes of all lysimeters across all combinations of parameters in the given ranges. Additionally, their interquartile range (IQR) is provided.

| Flux | Parameter range | Mean [mm] | Median [mm] | IQR [mm] |
|------|-----------------|-----------|-------------|----------|
| **Fog** | [95, 100] % | 6.7 | 6.7 | 10.4 |
| **Dew** | [-2, 0] °C | 9.4 | 8.0 | 7.1 |
| **Adsorption** | no perturbation | 24.2 | 22.8 | 5.3 |
| **Residuals** | - | 25.3 | 25.2 | 3.6 |

Plausibility of the partitioning parameter $rH_t$ for the distinction between fog and dew was tested in two steps. First conditions of thick fog were visually assessed from the digital camera imagery (exemplary shown in Fig. G1). During these conditions a median ratio of $LW\downarrow$ over $LW\uparrow$ of 0.98 was found (Fig. G2). During non-foggy conditions the median ratio is lower (0.88). The difference is statistically significant (p < 0.001). This shows that during fog $LW\uparrow$ and $LW\downarrow$ are close to equilibrium. The positive lysimeter $\Delta W$ assigned to fog based on $rH_t = 97\,\%$ are also closer to radiation equilibrium (median = 0.9) compared to dew (median = 0.84) (Fig. 5). The difference between the two categories is statistically significant (p < 0.001), despite the distributions overlapping. This applies also to $rH_t$ of $100\,\%$ or $95\,\%$ but the median radiation ratios during lysimeter classified fog conditions shift to 0.94 and 0.86, respectively (see Fig. G3).

To assess whether the thermodynamic requirements for dew and adsorption are met at our site we compare the lysimeter-inferred observations of dew and adsorption with their potential occurrence determined with the models from PM, equilibrium evaporation, and the aerodynamic diffusion equation (Fig. 6). Our results show that the measured fluxes are temporally consistent with model results, both concerning diurnal and seasonal dynamics.

In general, the Majadas site has suitable conditions for dew between October 2019 and end of May 2020, from sunset to sunrise. The statistical metrics for the comparison of daily dew duration between lysimeters and models are summarized in Table F1. They show that overall, the models suggest a longer duration of dew conditions by $3\,\mathrm{h\,day^{-1}}$ to $5\,\mathrm{h\,day^{-1}}$. Model statistics from PM and the evaporation model are not deviating from each other indicating that in our application, no difference between the simplified and full PM model is detectable. In the case of adsorption the lysimeter-based estimates agree better with the model predictions (Table F1). When comparing only measurements where at least two out of the five lysimeters show weight increases assigned to adsorption, mae and rmse decrease from $4.9\,\mathrm{hours\,day^1}$ to $4.0\,\mathrm{hours\,day^1}$ and from $5.9\,\mathrm{hours\,day^1}$ to

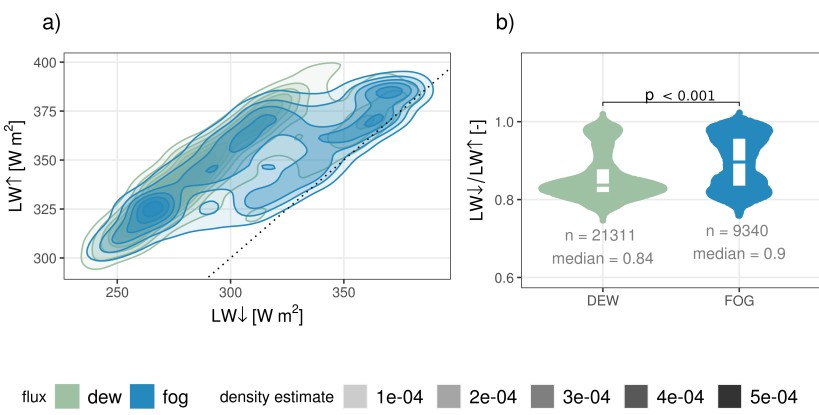

**Figure 5.** Comparison of upward ($\uparrow$) and downward ($\downarrow$) longwave radiation ($LW$) during the conditions classified as fog and dew from the flux partitioning of lysimeter weight changes. Panel a) illustrates the smoothed Kernel-Density-Estimate, with the black dotted line displaying the identity line. Panel b) illustrates the distribution of radiation ratios during dew and fog conditions. The difference between the distributions is statistically significant with a significance level p < 0.001.

$4.7\,\mathrm{hours\,day}^{1}$, respectively. In addition, the agreement of lysimeters is overall stronger during adsorption than during dew conditions (Fig. 6). Single lysimeters, however, frequently also measured adsorption until midday and before sunset.

## 4 Discussion

In this study, we showed that the climatic conditions in a semi-arid Mediterranean savanna site fulfill the thermodynamic requirements to induce diel cycles of evaporation and condensation at almost all times of the year. This finding is based on large weighing lysimeter and a few ancillary measurements. The routine applied to detect and distinguish NRW inputs is benchmarked with measurements of $LW$ and validated against models based on energy balance and moisture gradients, regarding the occurrence of the process.

### 4.1 Non-rainfall water frequency, duration and amounts

We found that NRW occur frequently at our site, in line with previous research in such a climate regime. The occurrence of both adsorption and dew was shown by Zhang et al. (2019b). We support their observation that dew formation and adsorption dominate at different times of the year. Regular dew formation ($120\,\mathrm{nights\,year}^{-1}$ to $200\,\mathrm{nights\,year}^{-1}$) has been reported across sites (Tomaszkiewicz et al., 2015). In a similar semi-arid steppe ecosystem in Spain, the mean number of days per year with suitable conditions for dew formation was $285\,\mathrm{days}$ (Uclés et al., 2014). However, this finding is based on measurements with micro-lysimeters consisting of PVC sampling cups with $< 0.1\,\mathrm{m}$ length, for which it has been shown that they tend to overestimate dew compared with alternative sampling approaches (Kidron and Lázaro, 2020; Kidron and Kronenfeld, 2020a,

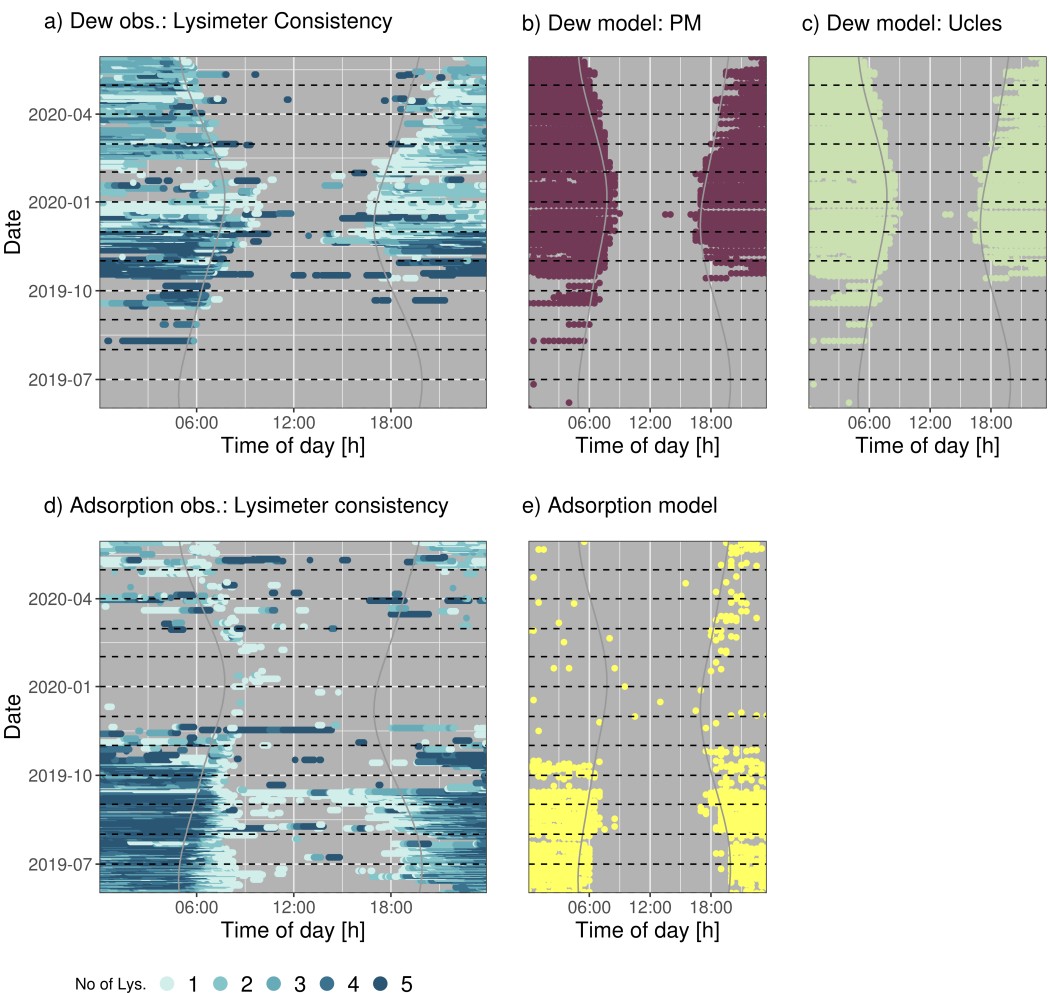

**Figure 6.** Diel and seasonal dynamics of consistency of inferred a) dew and d) adsorption, across all lysimeters. Modeled occurrence is presented for dew from b) Penman-Monteith model (PM), c) equilibrium evaporation model, and for adsorption using e) the aerodynamic diffusion equation. Solid lines mark sunrise and sunset, determined by the geographic coordinates of the site.

b). Our observation that especially nights are prone to the formation of NRW is also documented in the literature. Dew forma-
tion length has often been reported to correlate with the length of the night (Tomaszkiewicz et al., 2015) and was in another
Spanish site reported to last on average $9.3 \pm 3.2\,\mathrm{hours\,night^{-1}}$ (Uclés et al., 2014). In contrast, for adsorption, reported
observation times differ. Kosmas et al. (1998) observed the flux to occur mainly between 0:00 and 6:00 h and also Saaltink
et al. (2020) found suitable night-time conditions for adsorption through a reversed gradient of vapor concentration between
soil and atmosphere from lysimeter observations and confirmed it with a fully coupled numerical model. Yet, Verhoef et al.
(2006) found adsorption occurring during the afternoon and ceasing at night. Since changes in $T_s$ and their effects on the phase

equilibrium of water are one of the main controlling factors of adsorption, the different findings between studies could be related to site-specific timing of surface exposure to radiation.

Next to the diagnosed occurrences, also the NRW input amounts we determined are within the range of previously reported estimates for similar climate regimes. At our study site Majadas, the largest annual NRW contribution is adsorption with 21.2 mm year$^{-1}$. This value is relatively low, compared to sites close to the sea, where values of $81\,\mathrm{mm\,year^{-1}}$ (Saaltink et al., 2020), and 26 mm to 110 mm between April and October have been reported (Kosmas et al., 2001). But it matches well with observations from Qubaja et al. (2020) that measured annual adsorption of $14\,\mathrm{mm\,year^{-1}}$ using flux chambers in a semi-arid pine forest in Israel. Daily adsorption was reported to compensate for 25 % to 50 % of $ET$ in a Spanish olive orchard (Verhoef et al., 2006), and even 93 % of $ET$ in the Negev (Florentin and Agam, 2017). With a maximum compensation of 42 % per week, our findings are comparable to the observations in the Olive Orchard. However, it should be noted that the two cited studies covered time periods of only several days and therefore the variability of these percentages across the season is not known.

Furthermore, dew and fog quantified here at $14.4\,\mathrm{mm\,year^{-1}}$ and $6.6\,\mathrm{mm\,year^{-1}}$, respectively, are within the range reported from other sites which are depending on site $4\,\mathrm{mm\,year^{-1}}$ to $39\,\mathrm{mm\,year^{-1}}$ for dew (Tomaszkiewicz et al., 2015) and $1.3\,\mathrm{mm\,year^{-1}}$ to $50\,\mathrm{mm\,year^{-1}}$ for fog (Zhang et al., 2019b; Kidron and Starinsky, 2019). It is important to remark, however, that part of the large variability concerning the length of occurrence and condensation rates for NRW inputs could be related to biases of the measurement devices. Many former studies based on micro-lysimeters likely overestimated NRW inputs due to greater heat loss through the walls compared to the surrounding unperturbed soil Kidron and Kronenfeld (2020b). Despite the current version of large weighing lysimeters being designed with lower boundary control in order to prevent temperature deviations from the surrounding soil, the extent to which heat loss through their walls will affect NRW has yet to be evaluated at sites with suitable instrumentation, particularly since NRW is sensitive to deviations in $T_s$ (Yokoyama et al., 2021). Such a comparison was not possible in our instrumental setup because $T_s$ was derived from measurements in one location only, which was situated outside the lysimeters (Figure A1).

## 4.2 Impact of methodological uncertainty

We found differences in the absolute annual NRW sums between individual lysimeters that can be attributed either to (i) spatial variability (heterogeneity) in the soil and vegetation characteristics affecting the energy balance, or (ii) instrumental and methodological uncertainty. Particularly for the latter, external disturbances by wind or animals and internal disturbances such as data gaps, and data processing have been shown to alter results significantly (Schrader et al., 2013; Nolz et al., 2013). Developing processing routines for raw data from large weighing lysimeters has challenged researchers during the last decade (Schrader et al., 2013; Peters et al., 2014, 2016, 2017; Hannes et al., 2015). To assess the robustness of the processing we compared rain across lysimeters as rain is expected to be similar across nearby lysimeters and heterogeneous soils and vegetation. The variation of rain across lysimeters is only 5 %, which is of a similar magnitude reported in other studies (Hannes et al., 2015; Schneider et al., 2021). However, NRW sums often deviated between different measurement instruments and manual sampling (Kidron and Starinsky, 2019; Kidron et al., 2000). Unfortunately, quantitative validation of NRW sums was

not possible in our current study. Model simulations independently confirmed, that the conditions at night are suitable for NRW, although especially for dew the simulations showed that the potential for dew formation is generally longer than actual occurrence, measured with lysimeters.

Particular at our field site, the spatial heterogeneity of soil and vegetation characteristics can affect our results (Nair et al., 2019). In fact, dew and adsorption amounts are both reported to vary substantially with the surface cover type (Uclés et al.,
2016), soil exposure (Kosmas et al., 2001), shading hours (Uclés et al., 2015), distance from trees (Verhoef et al., 2006; Qubaja et al., 2020) and soil texture, particularly clay and sand content (Kosmas et al., 2001; Orchiston, 1953; Yamanaka and Yonetani, 1999).

Vegetation height and density affect the radiative exchange of the soil surface and the plant canopy affecting dew formation occurrence and amount (Monteith and Unsworth, 2013). (Xiao et al., 2009), for example, observed a positive relationship
between maize plant height and the amount of dew formed per night over two years. The grass height and LAI in Majadas change over the season reaching a maximum in height and density usually in winter and spring (Migliavacca et al., 2017). This is in line with our findings of the occurrence of dew and fog nearly exclusively during the wet season (although it is not possible to separate the effects from meteorological conditions). Unfortunately, no data on plant height and density were collected continuously in the lysimeters columns, which would have allowed studying their effect on the amount of dew
formation (and fog deposition). However, due to the active vegetation on the lysimeters, the effect of their variation in height and density is represented in our results.

Verhoef et al. (2006) showed in a measurement campaign with eight lysimeters concentrically arranged around a single oak tree that at the most exposed spots adsorption was doubled compared to the shaded spot. At our site, there are also individual sparse trees (Bogdanovich et al., 2021) which cause small-scale differences in shading. Since some lysimeter columns are
more exposed than others, part of the deviation in NRW input could be explained by spatial heterogeneity. This is supported by measurements of soil $\Psi$ within the individual columns, which showed that one column had an overall greater mean $T_{soil}$ in summer and the threshold of potential adsorption conditions was reached nearly a full month earlier than in other columns. Micrometeorological variables were however not measured individually at each column and therefore we have no insight into the exact causes of spatial heterogeneity in dew formation. The applied models both only suggest times of dew formation
potential based on measurements at the central facility, while the quantity is derived from weight changes. For the same reason, spatial differences are robust and follow-up investigations can be targeted towards understanding their causes.

### 4.3   Distinguishing non-rainfall water formation

Another focus of this study was partitioning the lysimeter $\Delta W$ into water flux classes. This approach includes the simplified assumption that one flux is always dominating over the others at each time step. In reality, the fluxes can occur simultaneously
with their relative importance shifting gradually over time (Li et al., 2021b). Ideally, we could account for a statistical probability ratio between different NRW inputs per time interval. But current research that is quantifying such ratios is too scarce for generalization (Li et al., 2021b). Some studies therefore don't distinguish between fog and dew (Groh et al., 2018) or add a category of combined dew and fog (Riedl et al., 2022). Disentangling individual NRW fluxes is nevertheless important because

of different respective i) controlling factors and ii) implications for the ecosystem. Better knowledge on controlling factors can help to identify potential NRW occurrence also in ecosystems without specialized measurement devices. The approach applied most frequently in the literature as well as in this study is based on a discontinuous tree-based classification system which was implemented similarly as suggested by Zhang et al. (2019a). We slightly deviated from their approach. The chosen temperature offset added to $T_{dew}$ was used in this study to account for the mean vegetation height. The soil surface, however, will exhibit warmer temperatures during the day. The height of the condensation surface should therefore always be considered.

We further added one node to account for prior knowledge on the controls of adsorption. The prior knowledge for soil adsorption stems, however, from smaller samples at equilibrium conditions in the laboratory (Arthur et al., 2016). Our application to point measurements of $rH$ and $SWC$ from above and below the soil surface assumes that by choosing the statistical upper envelope we can distinguish equilibrium conditions from remaining noise in the time series as best as possible. Although uncertainty remains about the actual shape of this relationship, this approach gives a more conservative estimate of the adsorption amount and helps prevent overestimation. The validity of this relationship is further confirmed by having a similar shape independently whether it was derived from the periods when lysimeter measurements unanimously were classified as adsorption (before including residuals as a final node) or deduced from negative EC derived $\lambda E$.

The advantage of the discontinuous decision tree-based classification is that it is applicable widely because the necessary data are commonly measured. The disadvantage is that selection of the parameters and thresholds in the classification algorithm is critical, especially at the upper nodes, where choices propagate into the estimates of flux classes of deeper nodes. Our test on different parameter combinations found that the strongest impact on flux quantity was for fog, followed by adsorption. Our results indicate that with a $rH_t$ of 97 % for the classification of fog layers, conditions that are opaque to $LW$ are predominantly recognized as fog. But there is also great overlap of the $LW$ ratio between the periods classified as dew and fog in moments where the surface cools and $LW\uparrow$ is larger than $LW\downarrow$. However, Feigenwinter et al. (2020) used visibilty sensors for fog detection in addition to micro-lysimeters and reported fog deposition continuing throughout several nights below the commonly used threshold of visibility $< 1000\,\text{m}$. The problem of distinguishing between dew and fog in particular has received great attention in the literature (Xiao et al., 2009; Price and Clark, 2014; Groh et al., 2018; Riedl et al., 2022) but practical difficulties concerning the distinction between dew and adsorption (Kidron and Kronenfeld, 2020a; Price and Clark, 2014; Feigenwinter et al., 2020), and fog and adsorption (Kidron and Kronenfeld, 2020a) were also mentioned. In our study, however, the ranking of the individual NRW contributions was not affected by the parameter thresholds, e.g. in all tested cases adsorption had the largest contribution to annual NRW input. We recommend future research to always test the effect of the chosen thresholds on the final NRW flux sums to have an uncertainty estimate of the chosen classification system.

Like other lysimeter studies, we assign dew flux as water input although no distinction between dew and soil distillation could be performed (Zhang et al., 2019b; Uclés et al., 2015; Riedl et al., 2022). Lysimeters readings, however, register the net water gain of the soil and plant in the monolith, and therefore mainly the input of external water (Nolz et al., 2014; Meissner et al., 2007). Because in theory, soil distillation should not affect the lysimeter net weight if the water vapor condensing on the leaf surfaces stems from the soil below in the same lysimeter (Li et al., 2021b). We support Li et al. (2021b) recommendation of combining isotopic composition measurements with lysimetric measurements in the future to verify this assumption.

## 4.4 Ecological relevance and open questions

An important finding of our study is that the relative share of adsorption to annual NRW input is with $50\%$ much larger than the contribution of dew. Since dew has received greater attention in the past (e.g. Tomaszkiewicz et al., 2015; Beysens, 2018) more long-term studies are necessary to evaluate which NRW flux is more relevant across years and semi-arid regions.

The role of dew has often been reported as moistening plant surfaces with direct leaf water uptake (Tomaszkiewicz et al., 2015). Our results show that in Majadas, dew occurs predominantly at a time of the year when top $SWC$ ranges between relatively wet values of 20 to $35\%$. Therefore, we assume that in this ecosystem dew as plant water supply is generally less relevant as has been reported for desert vegetation (Hill et al., 2015, e.g.). Dew may benefit plants indirectly for example by cooling the leaves during early summer or facilitating nutrient uptake over leaf tissues (Dawson and Goldsmith, 2018). Although we only investigated grassland, this could also be relevant for *Quercus ilex*, which is confirmed to allow water penetration from the upper leaf surface into the leaf interior (Fernández et al., 2014). As opposed to dew, soil vapor adsorption occurs at low $\Psi$ when grassland in Majadas has already senesced. Therefore, a potential ecological relevance would rather be to enhance microbial activity and trigger respiration from soil or (standing) litter (Evans et al., 2019; McHugh et al., 2015; Dirks et al., 2010; Li et al., 2021a; Gliksman et al., 2017). However, the highest $CO_2$ emissions during summer in terms of volume are expected to be caused by rain pulses (López-Ballesteros et al., 2016). Reports exist also on nighttime $CO_2$ uptake during adsorption but the underlying biogeochemical processes are not yet clear (Lopez-Canfin et al., 2022). Future research is necessary to disentangle and quantify the ecosystem response to the different types of NRW.

In our analysis we observed a similar temporal pattern between negative EC derived $\lambda E$ and lysimeter measured atmospheric vapor adsorption occurrence (Fig. 7). EC instruments are currently one of the most popular instruments to estimate $\lambda E$ at an ecosystem scale (Baldocchi, 2014, 2020). But measurements are frequently discarded when the underlying micrometeorological assumptions of the technology are not met (Göckede et al., 2004). This often affects night-time EC measurements (Massman and Lee, 2002).

Figure 7 underpins results from Florentin and Agam (2017) who compared $\lambda E$ fluxes from EC and micro-lysimeters in the Negev in the dry season and found, that EC was able to detect the dynamics but not the magnitude of adsorption from atmospheric humidity. Their study covered, however, only a period of seven days. Previous research in a pine forest in Israel also indicated that EC-derived $\lambda E$ tends to be negative at night during adsorption (Qubaja et al., 2020).

More research in paired lysimeter EC setups is necessary to scientifically assess the suitability and limitations of EC-derived night-time $\lambda E$ to detect and quantify adsorption of atmospheric vapor on soil material. If found suitable, EC instruments would help to spatially and temporally scale up NRW research. Additionally, they could help clarifying the role of NRW across research communities (Gerlein-Safdi, 2021), for example concerning energy balance closure of EC (de Roode et al., 2010), ecological significance of foliar water uptake (Berry et al., 2019) and impacts on remote sensing products (Xu et al., 2021). Irrespectively, our findings underline the necessity for methods measuring processes of the water cycle during the night to avoid biased measurements towards evaporation while missing condensation.

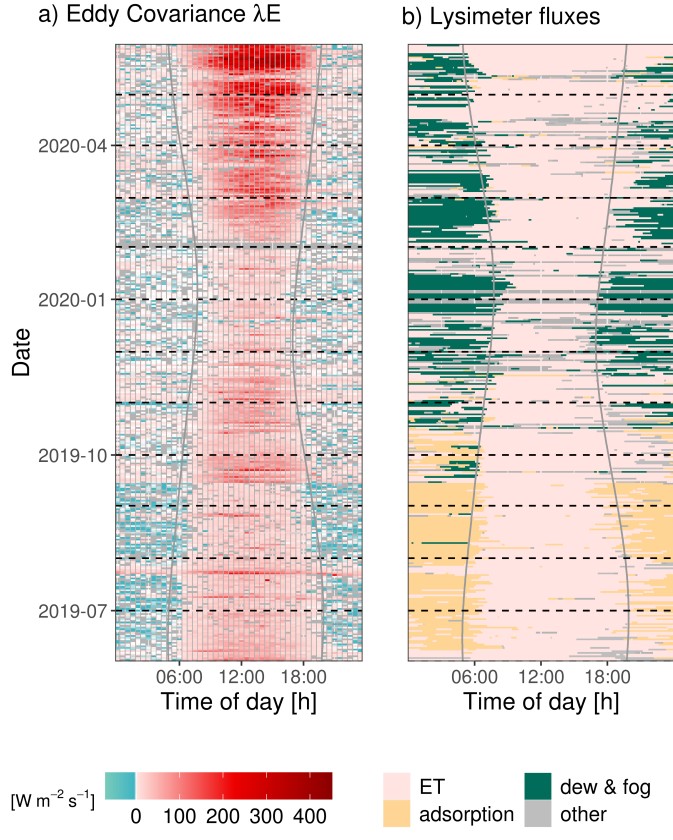

**Figure 7.** Diel and seasonal variation of a) quality filtered λE fluxes from EC and b) assigned lysimeter fluxes. Solid curved vertical lines mark sunrise and sunset, respectively, determined with the geographic coordinates of the field site.

## 5   Conclusions and outlook

In this manuscript, we derive NRW from time series of automated weighted lysimeters and compare their length of occurrence with established model estimations. In summary, our data suggest that this semi-arid savanna ecosystem switches between evaporation and condensation almost daily. Attributing the condensation pattern into different NRW fluxes sheds light on the distinct mechanisms that are each dominant in a different season. In summer, adsorption of atmospheric vapor on soil particles is facilitated by large diel temperature differences and dry soils lead to steep gradients of atmospheric vapor pressure between the atmosphere and the soil air driving vapor diffusion. In winter and spring, high $rH$ leads to frequent fog deposition and surface cooling to dew condensation. Spatial heterogeneity of vegetation, soil characteristics, and radiation regimes, together with measurement uncertainties are stronger reflected in NRW inputs than in other fluxes of surface water exchange. Although between 01.06.2019 and 31.05.2020, the total NRW input sum comprises only 7.4 % of the local water input, the relative con-

tribution strongly varies weekly. Rain frequency is unevenly distributed within the year and especially atmospheric adsorption stands out as the only water input during $11\,\mathrm{weeks}$ in the dry season. The ecological relevance of these fluxes has yet to be scrutinized. Until a quantitative validation has been carried out for NRW from large weighing lysimeters, the measured sums should be interpreted with caution. Ideally, such validation would take the form of several campaigns with manual sampling of soil and plants throughout the year to cover the seasonally varying NRW fluxes.

Our analysis focuses on lysimeters, that cover only a spatial area of $1\,\mathrm{m}^2$, each. We show that the temporal variability of the NRW derived from the instruments is coherent with negative $\lambda E$ fluxes at dry conditions. Based on this observation, future work could focus on revalue night-time $\lambda E$ fluxes from EC instruments to improve the spatial representativeness and assess the relevance of NRW at the larger scale and across seasonally dry ecosystems.

*Code and data availability.* Data and the R-code to reproduce the results of this analysis are available as Paulus et al. (2022) under https://doi.org/10.5281/zenodo.7354493.

*Author contributions.* MM, RO, MR, OPP, SP designed the setup and planned the study. OK, GM, AC, TEM, MM, and OPP were maintaining the site and instrumentation and conducted the measurements. SP, TEM, and TW processed the data. SP analyzed the data and prepared the original draft, both under the supervision of MM, RO, TEM, and AH. All authors discussed, reviewed and edited the manuscript.

*Competing interests.* The authors declare that they have no conflict of interest.

*Acknowledgements.* SP acknowledge the support for this research through the International Max Planck Research School for Global Biogeochemical Cycles (IMPRS-gBGC) at the University of Jena. The authors acknowledge the Alexander von Humboldt Foundation for supporting this research through the Max Planck Research Prize 2013 to Markus Reichstein. RO was supported by funding from the German Research Foundation (Emmy Noether Grant 391059971). We thank the city council of Majadas de Tietar for support. Our thanks go to Martin Strube, Ramón López-Jimenez, Martin Hertel, and the Freiland group at the MPI-BGC for great technical and scientific assistance during fieldwork. SP would further like to thank Andre Peters and Johanna Blöcher for their helpful comments and suggestions. We thank Prof. Dr. Werner Eugster, Dr. Giora J. Kidron and two anonymous reviewers for their valuable comments on earlier drafts of the manuscript.

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

# Appendix A: Lysimeter images

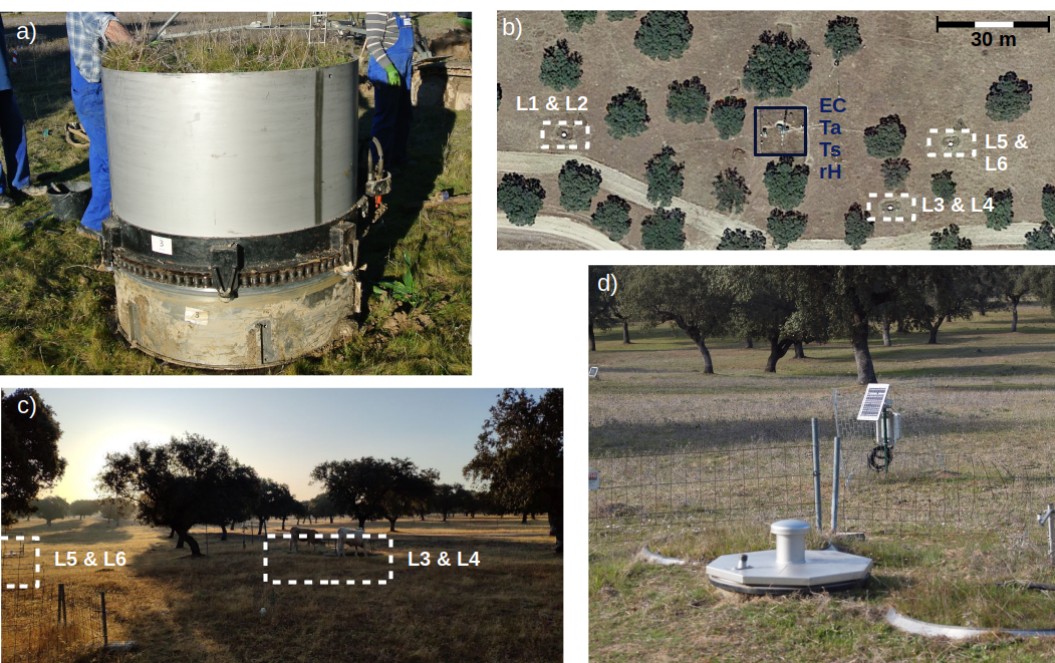

**Figure A1.** Pictures of a) intact, vegetated soil monolith in the lysimeter vessel after excavation for installation in 2015 (picture by Oscar Perez-Priego); b) aerial view of the measurement setup with the lysimeter stations marked by white rectangles, and the EC and meteorological sensors marked by dark blue rectangles (Map data from: Google Earth, Image Inst. Geogr. Nacional); c) exemplary image of the site on September, 26th at 8:30 in the morning to illustrate the heterogeneous shading conditions caused by the singular standing trees; d) lysimeter column 3 on March 03rd, 2022. The low fence around the station is installed to allow grazing of cows on the column, keeping them comparable with herbaceous vegetation outside the lysimeters, while reducing the risk of cows stepping on them (picture by Gerardo Moreno).

## Appendix B: Symbolslist

| Symbol | Full form | Unit |
|---|---|---|
| $C_p$ | Specific heat capacity of dry air = 1004 | $\mathrm{J\,kg^{-1}\,K^{-1}}$ |
| $G$ | Soil heat flux | $\mathrm{W\,m^{-2}}$ |
| $R_{net}$ | Net radiation | $\mathrm{W\,m^{-2}}$ |
| $SWC$ | Volumetric soil water content | $\mathrm{m^3\,m^{-3}}$ |
| $T$ | Temperature | °C |
| $T_a$ | Air temperature | °C |
| $T_s$ | Surface temperature | °C |
| $T_{dew}$ | Atmospheric dewpoint temperature | °C |
| $T_{soil}$ | Soil temperature | °C |
| $\gamma$ | Psychrometric constant | $\mathrm{kPa\,K^{-1}}$ |
| $\rho_a$ | Density of air | $\mathrm{kg\,m^{-3}}$ |
| $s$ | Derivative of the saturation vapor pressure curve defined as $(de_{sat}/dT)$ | $\mathrm{kPa\,K^{-1}}$ |
| $ET$ | Evapotranspiration | mm |
| $G$ | Ground heat flux | $\mathrm{W\,m^{-2}}$ |
| $LW$ | Long wave radiation | $\mathrm{W\,m^{-2}}$ |
| $SW$ | Short wave radiation | $\mathrm{W\,m^{-2}}$ |
| $\Delta W$ | Lysimeter weight change | $\mathrm{kg\,min^{-1}}$ |
| $\Delta q$ | Deficit in specific humidity at reference level | $\mathrm{kg\,kg^{-1}}$ |
| $\Psi$ | Soil matric potential | hPa |
| $\lambda E$ | Latent heat flux | $\mathrm{W\,m^{-2}}$ |
| $\sigma$ | Boltzmann's constant = $5.67 \times 10^{-8}$ | $\mathrm{W\,K^{-4}\,m^{-2}}$ |
| $\varepsilon$ | Emissivity of grass cover = 0.99 | (—) |
| $e_a$ | Actual vapor pressure of the atmosphere | kPa |
| $e_{s,0}$ | Vapor pressure of soil air at the soil surface | kPa |
| $e_{sat}$ | saturation vapor pressure (determined by temperature) | kPa |
| $rH$ | Relative humidity | % |
| $r_{av}$ | Aerodynamic resistance to vapor transport between the surface and air | $\mathrm{s\,m^{-1}}$ |
| $u_*$ | Friction velocity | $\mathrm{m\,s^{-1}}$ |
| $u$ | Wind speed | $\mathrm{m\,s^{-1}}$ |

## Appendix C: Additional equations

**Surface temperature** ($T_s$, °C) was calculated from measurements of the radiometric tower

$$T_s = \sqrt[4]{\frac{1}{\sigma \cdot \varepsilon} \cdot [LW \uparrow - (1 - \varepsilon)LW \downarrow]} - 273.15 \tag{C1}$$

where $LW$ is upwelling ($\uparrow$) and downwelling ($\downarrow$) long wave radiation ($\mathrm{W\,m^{-2}\,s^{-1}}$), $\sigma$ is Boltzmann's constant ($\mathrm{W\,K^{-4}\,m^{-2}}$) and $\varepsilon$ is emissivity of grass (—).

**Dewpoint temperature** ($T_{dew}$, °C) was calculated from $rH$ and $T_a$ based on the Magnus equation ($\lambda = 17.62$, $\beta = 243.12$)

(Sonntag, 1990):

$$T_{dew} = \frac{\lambda \cdot \left(ln\left(\frac{rH}{100}\right) + \frac{\beta \cdot T_a}{\lambda + T_a}\right)}{\beta - \left(ln\left(\frac{rH}{100}\right) + \frac{\beta \cdot T_a}{\lambda + T_a}\right)} \tag{C2}$$

where $rH$ is relative humidity (%) and $T_a$ is air temperature (°C).

**Equations for evaluation statistics** used to compare measured ($y$) and modeled ($\hat{y}$) duration lengths of dew and adsorption, respectively. $n$ is the number of observations and $m_y$, and $m_{\hat{y}}$ correspond to the means of $y$ and $\hat{y}$, respectively.

$$cor = \frac{\sum (y - m_y)(\hat{y} - m_{\hat{y}})}{\sqrt{\sum (y - m_y)^2 \sum (\hat{y} - m_{\hat{y}})^2}} \tag{C3}$$

$$mae = \frac{1}{n} \sum_{i=1}^{n} |\hat{y}_i - y_i)| \tag{C4}$$

$$rmse = \sqrt{\frac{1}{n} \sum_{i=1}^{n} (\hat{y}_i - y_i)^2} \tag{C5}$$

**Appendix D:  Example lysimeter weight evolution and flux categories**

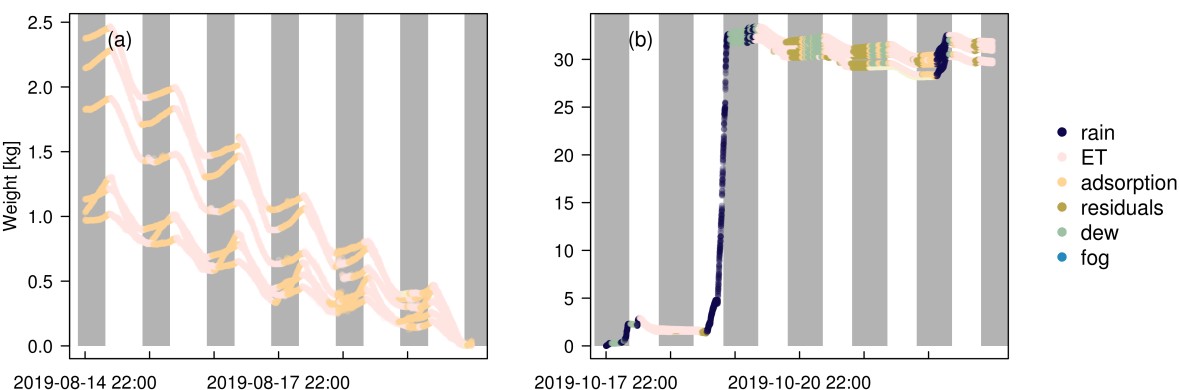

**Figure D1.** Lysimeter weights (normalised over the shown time period for each lysimeter by subtracting the respective minimum weight) exemplary shown for (a) a week without rain from August 15th. to August 20th., 2019, and (b) a week with several rain events from October 18th., to October 23th., 2019. The color code shows the respective flux that was assigned to the preceding weight change. Grey shaded areas illustrate nights.

**Appendix E: Monthly number of days with NRW**

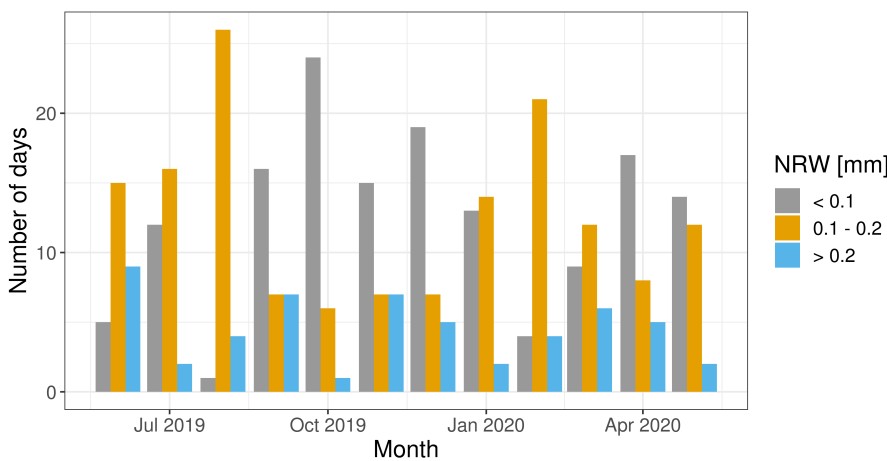

**Figure E1.** The number of days per month with average NRW input $< 0.1$, $0.1$ - $0.2$, and $> 0.2\,\mathrm{mm\,d^{-1}}$.

**Appendix F: Model comparison**

**Table F1.** Evaluation statistics for the comparison of the occurrence duration between lysimeter measurements and modeling. Dew was modeled based on Penman-Monteith (PM) and the equilibrium evaporation model model and adsorption based on the aerodynamic diffusion equation. Statistics are shown for the options i) more than one and ii) more than two lysimeters measuring dew or adsorption, respectively.

| Flux | Model | #n lysimeters | Statistics | | |
|---|---|---|---|---|---|
| | | | cor [-] | mae [h day$^{-1}$] | rmse [h day$^{-1}$] |
| **Dew** | PM | >1 | 0.44 | 3.32 | 4.76 |
| | | >2 | 0.36 | 3.78 | 5.30 |
| | Equilibrium evaporation | >1 | 0.44 | 3.32 | 4.76 |
| | | >2 | 0.36 | 3.78 | 5.30 |
| **Adsorption** | Aerodynamic diffusion equation | >1 | 0.76 | 4.90 | 5.8 6 |
| | | >2 | 0.76 | 4.00 | 4.70 |

## Appendix G: Phenocam

Suspended droplets during fog cause an optically thick layer that obstructs surface radiation to the sky and creates a radiation equilibrium. Under such conditions, visibility sensors would also record a decrease in visibility which has been used in other studies additionally to $rH$ sensors to separate fog from dew (e.g. Feigenwinter et al., 2020; Riedl et al., 2022). We visually identified mornings of fog and mornings where no fog occurred from images collected by a digital camera installed at the site from 01.10.2020 to 31.12.2020 between 6 am and 2:30 pm. A direct comparison between the camera images and the lysimeter measured fog deposition was not possible because fog could only be visually identified from pictures after sunrise. The deposition recorded with the lysimeters, however, stopped with sunrise. Lysimeter $\Delta W$ are examplary shown with measurements of $rH$ at different sensor heights and longwave radiation measurements in Fig. G1. Similar to measurements with a visibility sensor, fog should creates a detectable radiation equilibrium at our sensor at 1.6 m measurement height. The hypothesis was tested by comparing the ratio $LW\uparrow$ to $LW\downarrow$ between foggy and non-foggy conditions identified from the digital camera with the non-parametric Mann-Whitney U test.

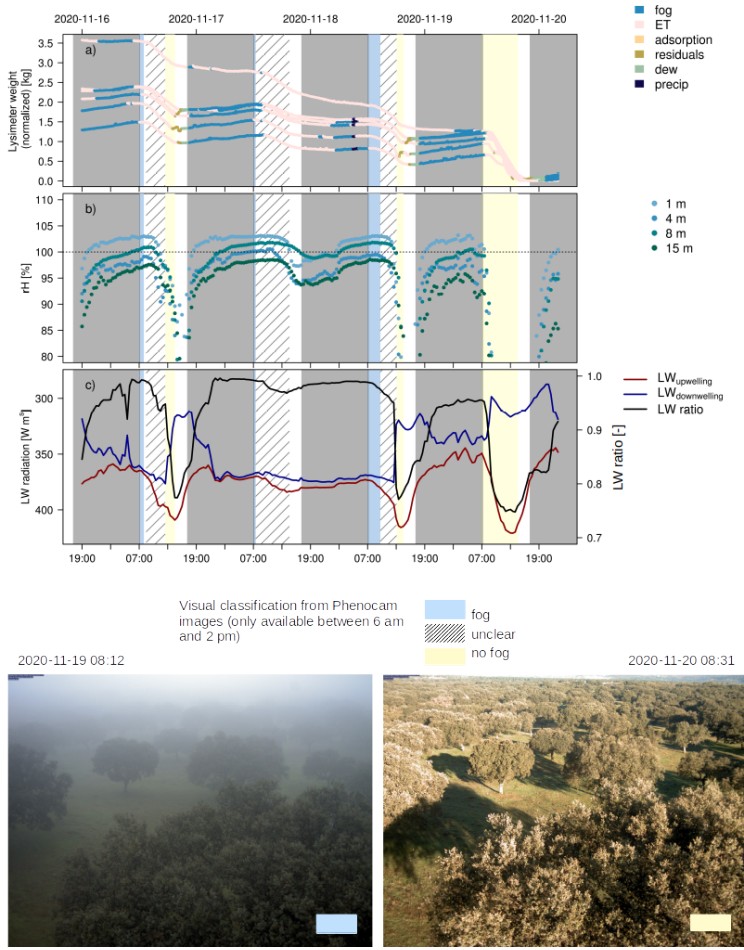

**Figure G1.** Timeseries of a) normalized weight changes of each lysimeter column with the respective assigned fluxes, b) $rH$ measurements and c) $LW$ upwelling and $LW$ downwelling and their respective ratios. Shaded areas illustrate nights (grey), conditions clearly identified as fog from digital camera images (blue, image bottom left) and non-foggy conditions (yellow, image bottom right). Hatched areas are used for periods that could not be unequivocally identified as foggy.

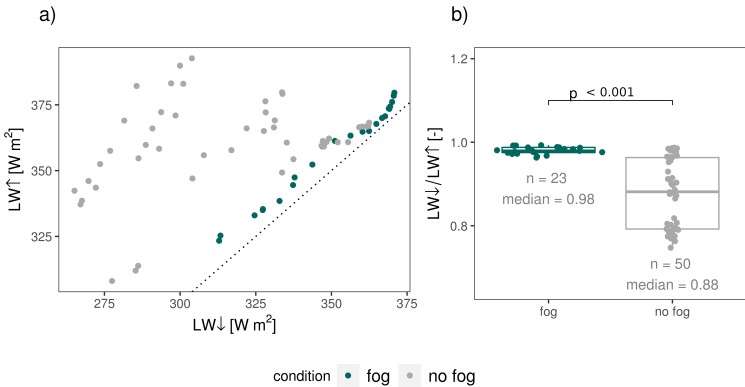

**Figure G2.** Comparison of upwelling $LW$ and downwelling $LW$ during the conditions classified as fog and without fog from a digital camera (example pictures shown in G1). The black dotted line in panel a) displays the identity line. Panel b) illustrates the radiation ratios during fog and no-fog conditions. The differences between the ratios are statistically significant with a significance level p < 0.001.

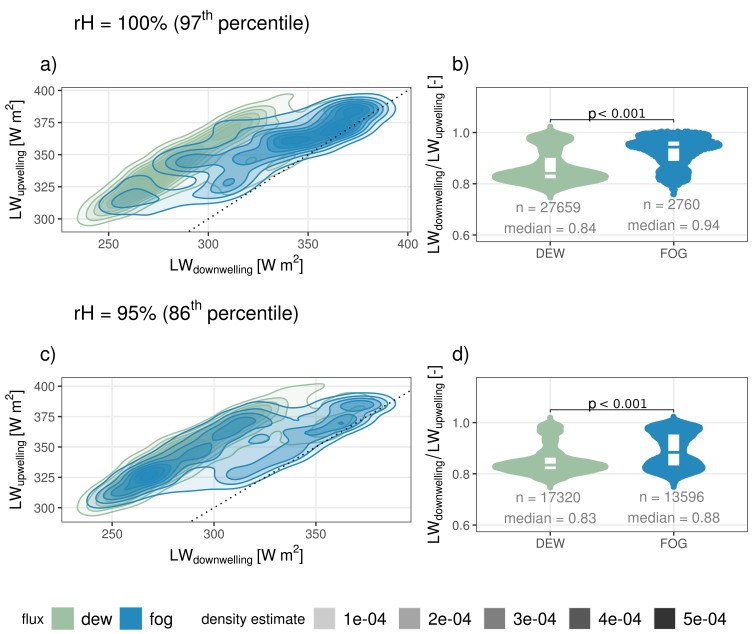

**Figure G3.** Comparison of upwelling $LW$ and downwelling $LW$ during the conditions classified as fog and dew from the flux partitioning of lysimeter weight changes based on a $rH$ threshold of $100\%$ (top row) and $95\%$ (bottom row). Panel a) and c) illustrate the smoothed Kernel-Density-Estimate, with the black dotted line displaying the identity line. Panel b) and d) illustrate the distribution of radiation ratios during dew and fog conditions. The differences between the distributions are statistically significant with a significance level p < 0.001.

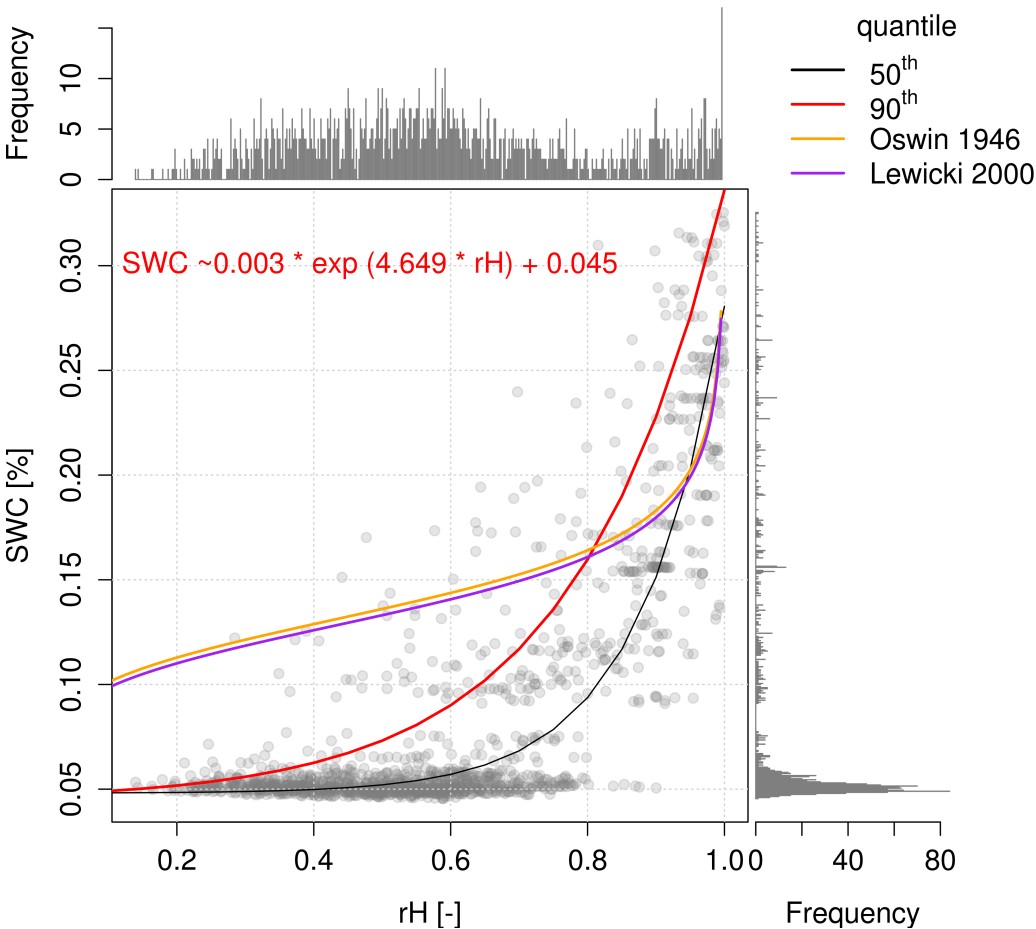

**Figure H1.** Relationship of $SWC$ and $rH$ during modeled adsorption (gradient driven vapor diffusion towards the soil surface) between April to October. The full formula with the parameters for the $90^{th}$ quantile from non-linear quantile regression is given in the upper left. The black line additionally illustrate the $50^{th}$ quantile. Empirical relationships from the literature for equilibrated conditions are shown in orange and purple (Oswin, 1946; Lewicki, 2000).