# Peer review of "Resolving seasonal and diel dynamics of non-rainfall water inputs in a Mediterranean ecosystem using lysimeters"

_Hydrology and Earth System Sciences, 2021_

## Author Comment (AC1)

| hess-2021-519 | Submitted on Oct 2021 |
|---|---|
| Lysimeter based evaporation and condensation dynamics in a Mediterranean ecosystem | |
| Handling Editor | Lixin Wang |
| Manuscript type | Research article |
| Status | Final response (HESS discussion) |

**Anwers to Referee #2 Giora Kidron**

**General comments**

The authors presented a thorough and clearly written research paper, which I enjoyed reading. There are some points, however, that should be pointed out, whether in order to clarify some issues or to point to possible directions for future research.

*Thank you very much for the comments and suggestions and for helping us to improve the manuscript. We considered them very helpful and changed the manuscript accordingly. Below, we provide answers to your comments (in italic).*

1. Unfortunately, the reader is not provided with a full description of the lysimeter. I assume that the lysimeter is covered with vegetation which reflects the vegetation at the site. Data regarding the cover and height of the vegetation at the site, along with photographs would have been helpful.

*Thank you very much for this comment, it is now clear to us that we should refer more clearly to the work of Perez-Priego et al. 2017, where a more detailed description of the setup is published. We added the following sentence to section 2.2 accordingly: "A full description of the technical details of the lysimeters is given in Perez-Priego et al. 2017."*

*We further followed your recommendation and added information on the vegetation in section 2.2 and added photographs of the setup and lysimeter cover in the Appendix (Figure A1):*

*"The lysimeters were installed in 2015 by excavating undisturbed soil monoliths from open grassland areas. The natural herbaceous vegetation cover was preserved. Pictures of the lysimeter columns and an aerial photograph of the site are shown in Fig. A1."*

[Figure]

[Figure]

[Figure]

[Figure]

*Figure 1. A1. Pictures of a) intact, vegetated soil monolith in the lysimeter vessel after excavation for installation in 2015 (picture by Oscar Perez-Priego); b) aerial view of the measurement setup with the lysimeter stations marked by white rectangles, and the EC and meteorological sensors marked by dark blue rectangles (Map data from: Google Earth, Image Inst. Geogr. Nacional); c) exemplary image of the site on 26.09.2019 at 8:30 in the morning to illustrate the heterogeneous shading conditions caused by the singular standing trees; d) lysimeter column 3 on March 03rd, 2022. The low fence around the station is installed to allow grazing of cows on the column during the grazing period while reducing the risk of cows stepping on them (picture by Gerardo Moreno).*

2. Scholars tend to refer to the amount of NRW that they obtained as representative of their site. Consequently, a comparison with other sites is often made, and similar values are taken as a supportive evidence for the reliability of the newly published data. Differences may be explained to stem from "site-specific timing" (I 302), but this may however not necessarily be the case. The authors were indeed aware of the complexity stemming from the sharp temperature gradient above ground, and the Td was corrected to that of 10 cm above ground. One however should note that this height above ground yielded the highest NRW in the Negev (Kidron, 1998, 2010). At higher height a decrease in NRW took place due to higher wind turbulence while at lower height a decrease in NRW stemmed from the heat emitted from the soil. While the 10 cm height taken by the authors is relevant for plants (as correctly indicated by the authors, and was in line with eddy covariance measurements), it is not necessarily relevant for biocrusts.

*If we understand correctly, the comment refers to the effect of the height of the vegetation on the amount of NRW (or dew in particular) and the effect of the sensor height to estimate if conditions are suitable for dew formation.*

*Concerning the amount of NRW it should not differ since in this setup the quantification is independent of height in contrast to the cloth plate or Plexiglas methods used in Kidron et al. (1998, 2010).*

*Concerning the reference height, we received a similar comment from referee # 3, Werner Eugster, on the approximated height. He suggested we should use an approximation for 1 cm height above the surface because this height was used in the work from Monteith 1957. As the reviewer points out here, the 10 cm height is relevant for vegetation at our site which has a mean canopy height of 0.1 m (Migliavacca et al. 2017). Biocrusts in contrast are rather rare which might be a consequence of regular grazing (Concostrina‑Zubiri et al. 2017).*

*Since we nevertheless understood that the manuscript is lacking more explicit reasoning for the readers we added some sentences on the effect of surface height on dew amounts and the relevance to take into account measurement height and the height of the condensation surfaces (vegetation or biocrust).*

*"Since the installation height of the sensor is 1 m we needed to approximate a value that better reflects $T_{dew}$ at the height of the condensation surfaces. Despite a reference height of 1 cm is generally used (Monteith, 1957), we approximated 10 cm which reflects the average canopy height of the herbaceous layer."*

3. Large lysimeters may better reflect the NRW. The extent to which heat loss through the walls of a large lysimeter will affect the NRW in comparison to the large effect recorded for microlysimeters (MLs) has yet to be evaluated.

*Yes, we agree. We are also not aware of a study that analyzes this effect. We added a sentence in the discussion to point out that in face of the identified overestimation of Micro-Lysimeters a similar analysis for large weighing lysimeters would be recommended before excluding the possibility of the same underlying effect (despite the lower boundary heat exchange system).*

*"Despite the lysimeters being built in a way that heat losses are prevented,  the extent to which heat loss through walls of a large lysimeter will affect NRW despite the lower boundary control has yet to be evaluated."*

Once MLs are used, as was the case for the Tabernas (Uclés et al., 2013, 2014, 2015, 2016), one may have assumed, based on the published NRW that the Tabernas may be considered as a 'dew desert' and that NRW has an important contribution to the biocrusts there. However, based on data from the Negev (Kidron and Kronenfeld, 2020a, 2020b; Kidron et al., 2021) and analysis of the microclimatological variables in the Tabernas (Kidron and Lázaro, 2020; Kidron and Kronenfeld, 2020c), the published data for the Tabernas should be taken with caution. It is not merely the distance from the Mediterranean (l 294), but rather the method used that may largely explain the differences in the reported NRW between the current site and the Tabernas.

*Thank you for this informative assessment. We changed the sentence in the following way:*

*"In a similar semi-arid steppe ecosystem in Spain, the mean number of days per year with suitable conditions for dew formation was 285 days (Uclés, 2014), however, this finding is based on measurements with micro-lysimeters and might therefore overestimate the real dew formation frequency (Kidron and Kronenfeld, 2020a, b)."*

4. Certainly, while the method employed by the authors may yield relatively reliable values in comparison to other methods which use lysimeters, **verification against manual measurements is necessary.** I assume that in this case, vapor condensation on the plant leaves should be measured. For a comparison to other sites (including the Tabernas) where great efforts were made to evaluate the amount of NRW obtained by biocrusts, direct NRW measurements also at the surface would be helpful.

*We fully agree with the reviewer that this would be most desirable and would also help identify whether the large weighing lysimeters potentially overestimate NRWI similar to Micro-lysimeters, as pointed out in comment 3.*

*However, since the start of this analysis, the traveling regulations were forcing us to cancel several field campaigns in which such data could have been collected. Therefore, the manual sampling was unfortunately not possible to perform and included in this analysis.*

*To point out this shortcoming of our study and that the NRW amounts should be handled carefully we extended the Discussion and the Outlook by the following sentences:*

*Discussion: "However, NRW sums often deviated between different measurement instruments and manual sampling (Kidron and Starinsky, 2019; Kidron et al., 2000). Unfortunately, quantitative validation of NRW sums was not possible in our current study."*

*Outlook: "Until a quantitative validation has been carried out for NRW from large weighing lysimeters, the measured sums should be interpreted with caution. Ideally, such validation would take the form of several campaigns with a manual sampling of soil and plants throughout the year to cover the seasonally varying NRW fluxes."*

***References***

*Concostrina‐Zubiri, L., Molla, I., Velizarova, E., & Branquinho, C. (2017). Grazing or not grazing: implications for ecosystem services provided by biocrusts in Mediterranean cork oak woodlands. Land Degradation & Development, 28(4), 1345-1353.*

*Kidron, G. J., Yair, A., & Danin, A. (2000). Dew variability within a small arid drainage basin in the Negev Highlands, Israel. Quarterly Journal of the Royal Meteorological Society, 126(562), 63-80.*

*Kidron, G. J., & Starinsky, A. (2019). Measurements and ecological implications of non‐rainfall water in desert ecosystems—A review. Ecohydrology, 12(6), e2121.*

*Kidron, G. J., & Lázaro, R. (2020). Are coastal deserts necessarily dew deserts? An example from the Tabernas Desert. Journal of Hydrology and Hydromechanics, 68(1), 19-27.*

*Kidron, G. J., & Kronenfeld, R. (2020a). Atmospheric humidity is unlikely to serve as an important water source for crustose soil lichens in the Tabernas Desert. Journal of Hydrology and Hydromechanics, 68(4), 359-367.*

Kidron, G. J., & Kronenfeld, R. (2020b). Microlysimeters overestimate the amount of non-rainfall water–an experimental approach. Catena, 194, 104691.

Monteith, J. L. (1957). Dew. Quarterly Journal of the Royal Meteorological Society, 83(357), 322-341.

Perez-Priego, O., El-Madany, T. S., Migliavacca, M., Kowalski, A. S., Jung, M., Carrara, A., ... & Reichstein, M. (2017). Evaluation of eddy covariance latent heat fluxes with independent lysimeter and sapflow estimates in a Mediterranean savannah ecosystem. Agricultural and Forest Meteorology, 236, 87-99.

Uclés, O., Villagarcía, L., Moro, M. J., Canton, Y., & Domingo, F. (2014). Role of dewfall in the water balance of a semiarid coastal steppe ecosystem. Hydrological Processes, 28(4), 2271-2280.

---

## Author Comment (AC2)

| hess-2021-519 | Submitted on Oct 2021 |
|---|---|
| Lysimeter based evaporation and condensation dynamics in a Mediterranean ecosystem | |
| Handling Editor | Lixin Wang |
| Manuscript type | Research article |
| Status | Final response (HESS discussion) |

**Author's response to Referee #3** Werner Eugster

(Referee comments in black, *author's response in blue italic*)

**General comments**

Non-rainfall water inputs (NRWI) can be an important hydrological water source to plants in arid and semi-arid ecosystems, but also elsewhere during dry spells and drought periods. The authors argue that so far most measurements were done with microlysimeters that may overestimate NRWI if their construction is of a simple type that does not attempt to bring microlysimeter soil temperatures in good agreement with the surrounding soil. Contrastingly, a normal size lysimeter---of which the authors have 6 on site, where one was excluded from the analysis---have the advantage that a temperature control of the soil slab is possible and hence less of the problems reported for microlysimeters should result from using standard lysimeters with highly resolving weight measurement.

The authors present a full year of data from a Mediterranean site in Spain, but the focus of the manuscript is more on the method, the real measurements are more used in a proof-of-concept mode without independent and reliable (and established) validation data as obtained from blotting paper water collection and analysis. Thus, the manuscript could actually be classified as a "technical note". My suggestion is to suggest moderate revisions before accepting the paper. There are a few scientific errors that can be easily rectified in a thorough revision round (wrong physical units, mostly) and with some information the context and wording can be quite misleading and should be corrected. Moreover, the main shortcoming of the manuscript is the

(1) the absence of a (at least simple) visibility sensor to be able to scientifically correctly separate fog from dew conditions, and

(2) lack of robust and independent validation data for supporting the claim that the presented method---which seems to be a further development of Zhang et al.'s (2019) method---is more accurate than other approaches.

Nevertheless, I recommend proceeding with the manuscript, there is indeed a need for better and more accurate quantification of NRWI in all ecosystems where rainfall can be absent for prolonged periods.

*We thank the reviewer for the constructive evaluation of our work and the detailed feedback. We do our best to improve our manuscript accordingly.*

*We would like to respond directly to the identified main shortcomings in the General comments. All other points will be addressed individually within the next section.*

*The reviewer identifies that the written manuscript has a strong focus on the methodological approach and suggests that the manuscript could also be classified as a 'technical note'. In our opinion, a technical note would require a complete validation including campaign-based measurements which we were not able to perform. This was criticized by reviewers 1 and 2, Giora Kidron and Werner Eugster, and we completely agree with both on this point. Unfortunately, however, we couldn't conduct such a campaign during the central measurement period, also due to restrictions related to the pandemic. Therefore, we alternatively tried to evaluate the lysimeter rainfall magnitude and coherence across stations and evapotranspiration measurement totals with independent observations and to benchmark NRWI as well as possible with models.*

*In the manuscript, we are nevertheless primarily interested in presenting the diel and seasonal dynamics of NRW at our site. Therefore, we prefer to keep the manuscript as a research article as it goes beyond the technical aspect. We leave it to the editor to decide whether he would classify this manuscript as a technical note.*

***Concerning (1)*** *based on the concerns of the reviewer we extended the analysis and tried to scientifically benchmark fog occurrence based on radiation equilibrium measured at 1.6 m sensor height. Suspended droplets during fog cause an optically thick layer that obstructs surface radiation to the sky and creates a radiation equilibrium. Under such conditions, visibility sensors would also record a decrease in visibility which has been used in other studies additionally to rH sensors to separate fog from dew (e.g. Feigenwinter et al. 2020, Riedl et al. 2022).*

***Methods:*** *(1) We visually identified mornings of fog and mornings where no fog occurred from images collected by a digital camera installed at the site from 01.10.2020 to 31.12.2020. (2) We hypothesize that fog creates a radiation equilibrium at our sensor at 1.6 m measurement height. We tested this hypothesis by comparing the ratio LW↑ to LW↓ between foggy and non-foggy conditions with the non-parametric Mann-Whitney U test.*

*(2) Then, we compare the ratio of LW↑ and LW↓ during fluxes classified as fog based on rH and the lysimeter weight changes to periods where dew or none of the two fluxes are assigned. Statistical significance between the conditions was again tested with the non-parametric Mann-Whitney U test.*

*A direct comparison between the camera images and the lysimeter measured fog deposition was not possible because fog could only be visually identified after sunrise. The deposition recorded with the lysimeters, however, stopped with sunrise.*

***Results1:*** *During fog conditions that were visually identified from the digital camera a radiation equilibrium is nearly reached with a median ratio of 0.98, whereas during non-foggy conditions the median ratio is lower (0.88). The difference is statistically significant (p < 0.001).*

[Figure]

*Figure 2. Comparison of upward (↑) and downward (↓) longwave radiation (LW) during the conditions classified as fog and without fog from a digital camera. The black dotted line in panel a) displays the identity line. Panel b) illustrates the radiation ratios during fog and no-fog conditions. The differences between the ratios are statistically significant with a significance level p < 0.001.*

**Results2:** *Lysimeter weight increases assigned to fog based on a rH threshold of 97 % are also closer to radiation equilibrium (median = 0.9) compared to dew (median = 0.84). The distribution of the ratios has a greater overlap at lower ratios. The difference between the two categories is statistically significant (p < 0.001).*

[Figure]

*Figure 3. Comparison of upward (↑) and downward (↓) longwave radiation (LW) during the conditions classified as fog and dew from the flux partitioning of lysimeter weight changes. Panel a) illustrates*

*the smoothed Kernel-Density-Estimate, with the black dotted line displaying the identity line. Panel b) illustrates the distribution of radiation ratios during dew and fog conditions. The difference between the distributions is statistically significant with a significance level p < 0.001.*

*In the manuscript, we changed the Material and Method section by reformulating 2.3 and adding a new subsubsection (2.3.1), explaining the approach. The respective results were added in section 3.2 as (new) Figure 5 (only for rH = 97 %). In the Discussion, we extend the paragraph (l 361 - 366) to put greater emphasis on the threshold parameter selection uncertainty and the link to the (partially) incorrect assumption of only one flux occurring at a time.*

*We hope that this approach convinces the reviewer that even without a visibility sensor a separation between fog and dew can be performed to a sufficient degree based on an rH threshold.*

**Concerning (2)** *we changed all passages in the manuscript claiming that the presented method is more accurate than the method of Zhang et al. 2019. As stated above, this was not the scope of the manuscript and cannot be performed without on-site measurements.*

**MAJOR POINTS**

Title: the paper is clearly focused on NRWI, and evaporation in my view is rather treated a side aspect (and not very clearly associated with NRWI except for the short discussion about downward latent heat flux being in good agreement with water vapor absorption estimates from the lysimeter). In my view a title of the kind of ``Lysimeter based quantification of non-rainfall water inputs to a Mediterranean ecosystem'' (maybe clearly classified as a technical note) would represent the contents much better.

*Thanks for sharing your thoughts on the title and for your concrete suggestion. Based on your concerns and our intention to formulate the core message of the paper as its title we would come up with the following suggestion as an alternative:*

*"Resolving seasonal and diel dynamics of non-rainfall water inputs in a Mediterranean ecosystem using lysimeters"*

62: ``Recently Kidron and Kronenfeld (2020b) found that temperature inside the micro-lysimeters deviated from that in the surrounding soil, \ldots''---here some rewriting is required. Firstly, the original paper makes unacceptable generalisations that require some caution when using that reference; secondly, the statement as presented here does not correctly reflect the contents. Kidron and Kronenfeld (2020b) used microlysimeters (ML) with a huge gap between ML and original soil, a cold-air trap that leads to excessive cooling of the pit at night and consequently to lower than normal soil temperatures. And that's the key effect, \textbf{not} that the soil temperature is different from the surrounding soil: if the soil inside the ML is colder than it should be, then you expect additional condensation to occur in the ML, thereby artificially suggesting and NRWI that is in fact an artifact. As you can see in Riedl et al. in Fig. 5 the soil inside our MLs is actually somewhat warmer or equal in temperature compared to the control. In this way, the artifact of the ML critisised by Kidron and Kronenfeld (2020b) is avoided. Thus, the problem with Kidron and Kronenfeld (2020b) is that they generalize from their overly simplistic ML to all ML (which is not correct, a simple lid

actually solves the problem, or at least reduces this artifact). A normal-size lysimeter with such a large gap around the lysimeter would also act as a cold air pit, the only advantage you have with a large lysimeter is that you could more easily add heating wires to heat the soil to more closely match the control temperature. Thus, I agree with your argumentation in lines 64--66.

*Thank you, we understand that this sentence was misleading in the way that it did not clearly point out that the problem lay in the ML design that many past studies were based on and not the ML technology itself. As suggested we reformulated the sentence to make this point more clear:*

*"It was recently suggested that the temperature regime in many formerly used micro-lysimeter setups deviated from the surrounding soil causing an overestimation of the measured NRW (Kidron and Kronenfeld, 2020b)."*

169: The 10\,cm $T\_a$ is not a real reference, see Monteith (1957). Why not extrapolate to the 1\,cm height? At least you must reference and consider Monteith (1957), this is an omission which is not understandable. In my view all papers with Monteith as author or co-author are of a quality that makes them relevant even after decades, and omitting Monteith knowledge normally goes in the wrong direction (scientifically), away from what we call ``progress''.

*Thank you for this comment. We included Monteith 1957 as a reference but would argue that the 0.1 m height of air temperature measurement is appropriately chosen based on the conditions at our site. The grass layer in Monteith 1957 was kept about 1 cm high by regular cutting, which is why this height is suitable for their setup. The average canopy height of grasses in Majadas is however 0.1 m (Migliavacca et al. 2017). Therefore, we prefer to keep the offset value for the temperature at 0.1 m height.*

*This choice has been indirectly confirmed by Giora Kidron (reviewer 2), who supported our choice for the grass layer, pointing out that may not be appropriate for biocrusts (which we barely have at our site).*

*Nevertheless, based on your comment we analyzed the data again and found that the average air temperature difference from 0.1 m measurement height and spline extrapolated temperature at 0.01 m height amounts to only 0.03 °C.*

313--315: see my remarks further up (line 62). This statement needs a rewording to take care of the flaw of Kidron \& Kronfeld's questionable generalisation to all ML, and the fact that this overestimation could be overcome easily by using a smarter ML design with a lid to avoid the nocturnal cold-air pit.

*We changed the sentence in lines 313-315 in the following way:*

*"Many former studies based on simple micro lysimeters likely overestimated NRWI due to greater heat loss through the walls compared to the surrounding unperturbed soil (Kidron and Kronenfeld, 2020b)."*

Appendix A1: this is never referenced in the text and is a mess---either remove or rectify all the errors. If the latter is desired: $C\_p$ has wrong units; $s$ see my comments; $LW$ and $SW$ have wrong units; $\Delta W$ has wrong units; use $\Delta q$ instead of $\delta q$;

$\varepsilon$ has wrong units (should be dimensionless or simply (---)); $e_a$ has wrong units; $u_*$ should have the asterisk in subscript

*Thank you very much for making us aware of these errors, we corrected them and added a sentence in section 2.1.2 referencing Appendix A1. We also carefully checked the rest of the manuscript to avoid other errors of this type.*

Appendix A2: Eq. (A1) would be simpler to read as \begin{displaymath}T_s = \sqrt[4]{\frac{1}{\sigma \cdot \varepsilon} \cdot \left[ LW_\uparrow - (1-\varepsilon) LW_\downarrow \right]} \end{displaymath}

*Thank you for making this suggestion, we made the changes to equation A1 as suggested.*

Appendix A2: the mathematical convention is to use either $\cdot$ (\verb+\cdot+) or space for multiplications of scalars, and only use $\times$ for vector products; please update equations accordingly. Use ``upwelling'' and ``downwelling'' for radiation fluxes; there is always the potential confusion that a down-looking sensor actually measures upwelling radiation, etc. Replace the erroneous NA with (---) or (dimensionless)

*Thank you for making this suggestion, we made the changes to equation A1 and in the methods section as suggested.*

Figure 6 and the text is actually a result of the study, not a discussion point. Maybe this is the reason why I think the title and text do not agree---if evaporation as mentioned in the title were the focus of the paper this aspect would have come first in the Results section, not last as an add-on in the Discussion section.

*We understand Figure 6 more like an outlook & hypothesis for future work. Changing it into a result of the study would require a more detailed and statistically substantiated analysis which does not fit the focus of the manuscript. Nevertheless, we think this observation is interesting to the community and promotes more research on EC negative latent heat flux measurements. We revised the respective paragraph and tried to make it more clear that future research should systematically test the suitability and limitations of EC to detect adsorption.*

*We would leave it to the editor to decide whether Fig. 6 should be moved to the results section.*

**OTHER IMPORTANT POINTS**

12: ``eddy covariance-derived latent heat flux estimates'': since the paper focuses on NRWI I found this statement somewhat misleading because it only addresses the ET losses but not the gains that would be associated with NRWI. Moreover, the authors do not even define ET, which indicates that this was not the real focus of the manuscript.

*We are not sure that we understand the problem with this sentence since we chose "latent heat flux" in the formulation for the reason that it is not directional, whilst ET as you point out here would only address the losses.*

*Also for the definition, we would like to ask if there is a problem with the way we defined ET*

*("evapotranspiration (ET, mm)") in line 18 or if this sentence was maybe just overlooked.*

23: horizontal precipitation does not belong to NRWI. It is rainfall (precipitation) which is not measured by standard rain gauges (but e.g. by special rain gauges on vessels).

*Thank you for clarifying this point. We deleted horizontal precipitation in the parenthesis.*

26: it is not Feigenwinter et al. who defined the classification of fog. Rather use a primary source here, AMS, WMO, or (what I use) the AMS Glossary of Meteorology by Glickman (2000):

```
@Book{Glickman2000,
  editor    = {Todd S. Glickman},
  publisher = {American Meteorological Society},
  title     = {Glossary of Meteorology},
  year      = {2000},
  address   = {Boston, MA},
  edition   = {2},
  comment   = {formerly: http://amsglossary.allenpress.com/glossary/},
  url       = {https://glossary.ametsoc.org/wiki/Welcome}
}
```

*We completely understand that applying this change makes great sense and exchanged the citation following the suggestions of the reviewer.*

30: ``However, differentiating between these two origins is commonly not possible (Li et al., 2021b)"---this is wrong, please read the text: Li just shows the opposite that because the sources of the vapor used in dew formation and the vapor from soil water are of such different origins, using stable isotopes allows to differentiate. You may argue about the word "common", but stable isotopes are common by now (at least the simple-to-measure ones such as $^{18}$O and $^2$H in water and water vapor). MPI Jena is doing this since its establishment. Please reword to convey the correct content and context with this statement.

*We agree that the citation of Li at this point in the text was misleading. What we want to express is that in most NRWI studies this differentiation was not performed and Li et al. 2021b were an exception. We changed the sentence in the following way and hope it is more clear now:*

*"However, most literature summarizes both processes as dew because a distinction requires additional measurements such as stable water isotope (Li et al. 2021b)."*

88: ``However, the structure of the cage allowed for grazing to maintain the lysimeters comparable with the rest of the plot."---this is a challenge and it would be interesting to read some more details how this is successfully done. As is, it is not possible to reproduce this as a reader.

*We added a sentence on the height of the fence (50 cm, cage was the wrong wording) and also referred to the pictures added in the appendix based on the suggestion of Giora Kidron (Reviewer 2). We hope that these additions make it more comprehensible for the reader.*

*For the period analyzed in the manuscript, the strong agreement between EC and lysimeter ET suggests that the grass cover inside the lysimeters was representative of the footprint of the tower in the open space.*

*During spring 2017, dry matter of the herbaceous vegetation was measured in-situ manually (GrassMaster Pro Drymatter Instrument, Novel ways Limited, New Zealand) on the lysimeter and open space. The mean drymatter over spring in the open area amounted to 704 kg/ha and to 714 kg/ha on the lysimeters. The difference of 10 kg/ha was however smaller than the standard deviation of 348 kg/ha for the open space and not systematically low, which would be the consequence when lysimeters were excluded from grazing. We could include this information of 2017 in the manuscript if the reviewer considers it an important addition.*

128--129: Note to Editor: I cannot check the data and code which will be made available once the manuscript is the same. I normally also only publish code and data once a manuscript is accepted, but here the authors do not provide the details in the paper and expect readers to go into the code.

*With submission, we sent the access links to the zenodo repository to the editor. The idea was that the data access would be thereby possible for reviewers that prefer remaining anonymous, and to then open up the database before acceptance.*

*If this didn't work out you can send us an access request on the following page and we grant the access to the data: https://zenodo.org/record/5575521*

*Apologies for the inconvenience.*

134: question: an animal stepping onto the column, is this not bringing $\Delta W$ outside of the accepted range and is thus treated there?

*If a large animal, such as a cow, is stepping on the column, this should have been removed by the fix threshold value (l 131). In line 134 we refer to lighter animals, such as rabbits or snakes, which should be identified by the comparison between lysimeter weight increase. We added this information in the text to make it more clear to the reader.*

*"In contrast, if only one lysimeter column shows an anomalous ΔW we considered this as an artifact (e.g., small animals such as snakes or rabbits stepping on the column or issues with the boundary control) that can be removed from the time series (Hannes et al. 2015).*

148: ``water input''---reword, NRWI is also a water input and this is \textbf{not} to be included here!

*We followed your suggestion and changed "water input" to "weight gain". We hope it is more clear now that we suspect a technical problem in these periods and do not consider these measurements reflecting surface-atmosphere exchange processes.*

166: That's why Monteith (1957) uses 1\,cm $T_a$ would be good if you could relate your text more to Monteith's outstanding work which is still our reference.

*We hope that we could address this point to the satisfaction of all by our answer in the section "Major points".*

251: you show ET with negative sign convention, although you use a positive sign convention for $\lambda E$ in Eq. (2.1). Moreover, you never defined ET nor its sign convention. My recommendation is to use positive values as ET losses from the soil to the atmosphere. I have not seen papers using the reverse sign convention, yours is the first, and this confuses me and maybe also the reader. Recall that the standard hydrological budget equation is still: precipitation = runoff + ET +/- change in soil storage. Moreover, you use a positive ET in Fig. 3b. At least you need to be consistent and declare your symbols and sign conventions (if they should deviate from common sense notation)

*We followed the recommendation of the reviewer and used positive values when referring to measured ET in the manuscript.*

280: ``evaluation statistics improve by one hour''---I am unable to see this one-hour improvement in Table A2. Is there an error here, either in Table A2 or in the wording?

*We checked again and there was no error in Table A2. We rephrased the sentence in the following way to be more specific:*

*"When comparing only measurements where at least two out of the five lysimeters show weight increases assigned to adsorption, MAE and RMSE decrease by from 4.9 hours to 4.0 hours, and from 5.9 hours to 4.7 hours, respectively."*

306: the absolute value of adsorption may be low (as total NRWI may be low), but the relative share in NRWI is quite high in my view. Maybe rather compare the relative numbers to focus on the processes leading to the different components of NRWI

*If we understood this comment correctly, in your view the relative share of adsorption to the total NRWI in our ecosystem is quite high. As suggested we added a sentence in the result section to include the information on the relative numbers:*

*"The largest relative contribution to the mean total NRWI is adsorption with 50 %, followed by dew with 33 % and fog with 15 %"*

*We also added the following sentence in the Discussion section to focus on the process:*

*"An important finding of our study is that the relative share of adsorption to annual NRWI is with 50 % much larger than the contribution of dew. Since dew has received greater attention in the past (e.g.Tomaszkiewicz et al., 2015; Beysens, 2018) more long-term studies are necessary to evaluate which NRWI flux is more relevant across years and semi-arid regions."*

*We also added references to other studies reporting on the relative share of adsorption to ET to underline that in this regard the relative share is reasonable and to underline the potential importance of its contribution to ET:*

*"Daily adsorption was reported to compensate for 25 % to 50 % of ET in a Spanish olive orchard (Verhoef et al., 2006), and even 93 % of ET in the Negev (Florentin and Agam, 2017). With a maximum compensation of 42 % per week, our findings are comparable to the observations in the Olive Orchard. However, it should be noted that the two cited studies*

*covered time periods of only several days and therefore the variability of these percentages across the season is not known."*

377: ``At our site, this pattern is obvious and indicates that night-time EC measurements could serve to detect adsorption (Fig. 6).''---I partially disagree, but you may convince me with your arguments. In my view adsorption is not directional as dew formation (from vapor above the canopy) or distillation (from vapor below the canopy), and hence should in my view not leave the best trace in EC-based flux measurements. I though (so far) that dew formation should lead to negative $\lambda E$, whereas distillation is not seen by an EC system; and adsorption should only be seen in EC fluxes if its vapor source is above the canopy, but not below. Your Figure 6 of course empirically shows better performance (qualitatively) for vapor adsorption than dew formation, but for me the explanation is not that obvious as your text implies.

*Thank you for sharing your thoughts and understanding of the process with us. It is important to differentiate between the processes i) adsorption of redistributed vapor within the soil column, and ii) adsorption of atmospheric vapor. This differentiation is not much different from soil distillation vs. dew. We agree EC should register negative $\lambda E$ when the $H_2O$ recorded from the lysimeters stems from the atmosphere (in the case of dew and adsorption of atmospheric vapor).*

*Our setup leads to the conclusion that the recorded adsorption mainly stems from the atmosphere and is not redistributed water within the soil column. This was also backed up (data not shown) by measurements of soil water potential from pF meters within the lysimeters at -10 cm depth, from which we can estimate soil pore vapor pressure (with the Kelvin equation). They revealed that even at 10 cm soil depth, pore vapor pressure drops occasionally below the atmospheric vapor pressure measured at 1 m height, particularly at night. Since close to the surface, the soil likely is even dryer, those observations pointed in the direction that the largest fraction of the adsorbed water at our site stems from the atmosphere.*

*But still - the question remains if EC and lysimeter adsorption measurements are comparable also in other setups and under different conditions with a different tower or vegetation heights. These questions also motivated us for a follow-up analysis where we started working on a set of paired lysimeter and EC observations from several sites, to systematically investigate these patterns across setups and hopefully can give you more profound answers after our next study.*

397: ``LE fluxes at dry conditions''---you never defined or used LE, but this appears to be an important statement that should have appeared in Discussion already. Conclusions should not bring up new aspects that were neither addressed in Results nor in Discussion.

*Thank you for pointing this out, we changed LE to λE, which was the term that we intended to use, in line with the figure and the reasoning presented in the Discussion. λE was defined in line 120 in the methods section already.*

Figure 1: in the text it sounds as if you want to use this as a general workflow also for other sites. But then my recommendation is to avoid site-specific magic numbers in the scheme and provide the site-specific values in the caption to be clear. E.g.: $T\_s < (T\_\textrm{dew} - T\_\textrm{dew, t})$ with the information that $T\_\textrm{dew, t}$ = 1.4\,$^\circ$C for your lysimeters and site.

*We changed Fig. 1 and the respective sentences in the methods section based on your suggestions.*

Figure 1: just of curiosity because we were challenged on this aspect with our ML: why do you not use the high-quality measurements of the drainage outflow of your lysimeter to make sure that drainage loss at the bottom of the soil slab is not erroneously treated as ET? You classify $\Delta W \le 0$ as ET, although it could be drainage loss after intensive rain. This may be an important aspect if you think the method should also be applicable elsewhere.

*We removed the tank level changes induced by the lower boundary control system (drainage and capillary rise) already in the raw data filtering, as described in line 130 The reason is, that in the step after - 'outlier identification across columns' - we needed to have these measurement system-induced changes already taken into account and excluded as a potential source of asynchronous behavior between columns.*

Figure 1: the percentile approach to separate fog from dew is in my view weak and questionable. Namely dew can only form under conditions that you classify as ``fog'' but fog droplets---if advected---can be present at relative humidities that are not showing saturated air (high $\mathrm{rH}$).

*It became clear to us that we need to explain better why we chose the percentile approach and what preconditions need to be fulfilled for this approach. We restructured the text and added to 2.2.1 (Material and Methods Section) the following information*

*"Theoretically, fog occurs at a rH of 100 %. We noticed, however, that the maximum saturation values varied depending on the sensor, with values between 98 % to 104 % (Fig. A2). We, therefore, decided to set a rH threshold ($rH_t$) that is based on the data distribution of the sensor to account for the individual uncertainty when the air is nearly saturated, for systematic biases, and for drifts. In our study $\Delta W$ is attributed to fog when $rH_t$ = 97.1 % which is the 90[th] percentile of the rH sensor records measured at 1 m height."*

*We added the sensor height to Figure 1 to make it also more clear that for fog, we look at sensors at 1 m height and for dew, the conditions within the canopy are considered.*

*At our site, advection fog is negligible which is reflected in low wind speeds and no constant wind direction during fog events.*

Figure 4: move the text in the dial at upper right so that there are no overlaps

Figure 4: reduce the size of the end marks of the whiskers

*Thanks for the nice suggestions. We implemented both suggestions in Figure 4.*

Figure A1: there is an error on the x-axis, this is $\mathrm{rH}$ as a fraction, not in \%. Moreover Oswin (1946) and Lewicki (200?) are missing in references. And please don't chomp off the right part of the display. I also cannot see the 95$^\textrm{th}$ percentile in my printout, and the 85$^\textrm{th}$ would profit from a thicker line width. Why do you use x and y in the equation when your variables are actually $\mathrm{rH}$ and $\mathrm{SWC}$? You never defined that x = $\mathrm{rH}$ and y = $\mathrm{SWC}$.

*We added the missing references and applied all suggestions to improve Figure A1. We identified that the 95$^{th}$ percentile could not be computed despite trying several starting estimates and decided to take out the lines illustrating the 85$^{th}$ and 95$^{th}$ percentile to stay consistent.*

Table A2: The caption claims that the units of all values are hours day$^{-1}$, but the table heading claims that this is only hours; and then there might be an error: if cor means correlation, a unitless and dimensionless information, both are not correct. To be standalone you must define cor, mae and rmse (you don't use mse defined on the previous page). Moreover, I cannot see what you want me to see according to the text (see comment elsewhere)

*We changed the equations in A2 to complete the definition of cor, mae and rmse and added the units individually to each column in table A2.*

**DETAILS**

32: Meissner et al. is not a complete reference, year etc. are missing

*Thank you for pointing this out, we corrected it in the manuscript.*

42: what do you mean with ``modeling frequency''---not intelligible to me

*We understand how the wording of the sentence was clumsy and now changed it to make it more clear:*

*"Modeling NRWI also remains a challenge, although a distinction must be made between modeling the frequency and duration of occurrence, and modeling the yields of the different NRWI fluxes."*

51: please add Riedl et al., accepted on 16 November 2021 at HESS. Last manuscript version available here (note that one more co-author appears in the finally accepted version): http://homepage.usys.ethz.ch/eugsterw/publications/pdf/Riedl.2021.FINAL.pdf, discussion paper version available via https://doi.org/10.5194/hess-2021-317

*We added the reference as suggested.*

77 and elsewhere: data are plural in formal English, please correct

*Thank you for pointing out this language-related error, we changed the respective sentences in the whole manuscript*

81: \textit{ilex} should be lower case; and remove the excessive white space before 20 trees

*We changed ilex to lower case, and identified the reason for the white space, there was a problem with the latex command to insert a tilde.*

95--96: ``They rest \ldots'': this sentence is not intelligible to me, please rephrase in an understandable way

*We rephrased the sentence to:*

*"Each column has a 1 m² surface area and 1.20 m column depth and is situated on a weighing system consisting of three precision shear-stress cells, respectively (Model 3510, Stainless Steel Shear Beam Load Cell, VPG Transducers, Heilbronn, Germany)."*

112: add ``and'' (Pt-100 and capacitive \ldots)

*We changed the sentence as suggested.*

115: add ``The'' in The Netherlands

*We changed the sentence as suggested.*

120 and elsewhere: $u_*$ always has the asterisk in subscript, never in superscript

*We made the change to the variable as suggested.*

120: add reduced space between $m$ and $s$

*In the original latex version of the manuscript we use consequently the siunitx-package, the spacing is done within the command and we would prefer to not change this for individual units.*

121: add a comma in R3-50, Gill \ldots

*We added the comma as suggested.*

127 and elsewhere: note that Figure should be upper case if a specific figure of your manuscript is referenced; it is however lower case if you use figure for ``Zahl'' or ``Wert'' in German

*Thank you for pointing this out as well as the clarification. We changed figure to Fig. as specified in the guidelines from hess.*

131: delete ``together''

*Was changed accordingly.*

139: not a number is \textbf{NaN}, whereas \textbf{NA} means not available (it is the code for missing values). That's wrong here. I assume you mean not available \textbf{NA}.

*Yes indeed, we mean not available. We changed the expression accordingly.*

184: add s to describe\textbf{s}

*Was changed accordingly.*

190: remove s from model predictions (not model\textbf{s} \ldots)

*Was changed accordingly.*

207: $\delta$ is conventionally only used for isotopic ratios, $\partial$ is used for partial differentials, and $\Delta$ is used for finite differences. Here I think using $\Delta q$ instead of $\delta q$ would reduce confusion if readers are familiar with isotopes, and personally, I even think that using capital delta is the correct notation anyway.

*Was changed accordingly.*

209: the Clausius-Clapeyron relationship is not a straight line and thus ``slope'' is the wrong word here. $s$ is actually $de/dT$; thus rewording is required

*We reworded and hope that the wording is more accurate now.*

*"s (Pa K⁻¹) is the derivative of the saturation vapor pressure curve defined as ( de $_{sat}$/d T)"*

209: there is an error here, $\lamda E$ is not the latent heat (of whatever), but the latent heat \textbf{flux}. Please correct.

*Was changed accordingly.*

221: you defined $\gamma$ to be in units of Pa K$^{-1}$, but here the implicit assumption is that is is in kPa K$^{-1}$. Stick to your definitions; if readers use the equation as is with saturation pressures in kPa, then the first term is 1000 times too large.

*Thank you very much for looking deeper into this, I was indeed mistaken here. We changed it in the finalized manuscript.*

223: $C_p$ is not specific heat of air. It is the specific heat \textbf{capacity} of the air at constant pressure. Please correct.

*Was changed accordingly.*

227: replace moments with periods

*Was changed accordingly.*

229: in the text you correctly use ``diel'', probably being aware that diurnal can also express the opposite of nocturnal. My suggestion is to modify the subsection title to match the text (diel)

*Thank you, you are totally right, we changed the section title accordingly.*

247: should be ``its'' without apostrophe

*Was changed accordingly.*

254: a sum has no $\pm$ unless you specify in M\&M how you obtained the uncertainty (the reason: random errors of the mean have an average of zero, and thus for a sum there are no degrees of freedom to specify a random uncertainty). Please correct.

*We added a sentence in section 2.2.1 to clarify the origin of the uncertainty.*

*"Fluxes are presented in the results section as mean and standard deviation across the five lysimeter columns."*

258: you arbitrarily change from ET to $ET$ -- please homogenise (and define the version you keep)

*Thank you for this suggestion, we revised the text again for homogeneous use of ET.*

263: your total of the components is 41.9 mm, but you specify 42.0 mm. The convention is to either specify the component that contains the missing 0.1 mm or to round accordingly if no additional component is part of the game. Note that modern round rules round 0.05 to 0.00 but 0.15 to 0.20 (this is essential to avoid drift). Another conventional rule is to round up the component that was closest to rounding down, to correctly represent the reported total.

*Thank you for having found this error. It should indeed be 41.9 mm in the text. We corrected it accordingly.*

264: I am surprised to see 50.6\% vapor adsorption. This seems to be quite a large value, but maybe is correct in this ecosystem. Here some independent (e.g. blotting paper) validation of the components would really have strengthened the paper.

*Yes, we agree with you regarding the fraction of adsorption from the total NRWI sum. This is one of the reasons why we presented this manuscript since little literature exists on periods of the length of a year which limits such comparisons. Of course, these values still could have a substantial interannual variability.*

*Regarding the field campaigns for validation measurements, we also agree. Unfortunately, nearly all field trips and potential campaigns in 2020 were canceled due to the Pandemic restrictions. But for the scope of the PhD project, it is crucial to continue now with the next step of the project.*

294: ``Our observation that especially nights 295 are prone to the formation of NRWI is also documented in the literature. '' This is an utterly trivial statement, do you really want to keep this in a scientific manuscript? For me it is on the same level as ``the grass was green and photosynthesis was important during daylight hours, as reported in the literature'' \ldots

*Yes, it is correct that nighttime dew formation is common knowledge. But still, we consider the sentence here as a way to guide the reader into the next sentences where we explain that particularly for adsorption, the observed timing differs from site to site (l.299). Some authors also reported nighttime adsorption while others observed the strongest flux between noon and sunset (Qubaja et al. 2020, Verhoef et al. 2006).*

336: should be ``its'' without apostrophe

*Was changed accordingly.*

379: use ``scale up'' instead of ``up scaling''

*Was changed accordingly.*

Figure 3: suggestion to move the legend to the panels and only show the curves that relate to the respective panel. In the legend some colors are hard to distinguish as is. Moreover, having a legend outside the plot area is Excel standard, not with scientific presentations.

Figure 3: use \verb+par(lend=1)+ to avoid the rounded (and thus unclear) endings of the bars

*We included all suggested changes in Figure 3.*

Figure 5: panel (e) is not described. I assume that the second mentioning of (d) should actually be (e)

*Thank you for pointing out this error. We changed d) to e) as you suggested.*

General: \LaTeX typesets equation by assuming that characters are variables (if they are known), hence $\mathrm{rH}$ and $\mathrm{SWC}$ look odd in your text. Consider using \verb+\mathrm{rH}+ and \verb+\mathrm{SWC}+ instead

*Our reasoning for this decision to have rH and SWC in italic is that we tried to consistently write in italic all symbols representing physical quantities or variables. They are defined within the LaTeX document in the glossary file and listed in the manuscript with their respective unit in A1. We would prefer therefore to keep them as they currently appear in the manuscript.*

References: add doi or URL to Thom et al. (with scanned papers the doi is normally only shown on the publisher's website)

References: add doi to Sonntag et al.

*Unfortunately, there is no doi available.*

References: add space before parenthesis in Zhang et al. (2019a)

References: check Dirks et al.

References: check Kosmas et al., seems to be an incomplete / corrupted entry

References: check Nair et al.

References: check Peters et al. (2014)

References: check Rodrigues-Iturbe et al.

References: generally only the doi is necessary, not doi resolved by the standard doi resolver plus the doi with the publisher's doi resolver and/or an alternative URL. See https://www.hydrology-and-earth-system-sciences.net/submission.html#references

References: generally rectify the entries according to the guidelines. Paper titles are normally in sentence case whereas journal names and book titles are using capitalised words

*Thank you for pointing this out, we used the Bibtex Bibliographic Style File from the hess submission guidelines. Theoretically, the style file would ensure that only the entries relevant for the standard hess reference list from the \*.bib file are considered.*

*As you identified there seems to be a problem concerning the URL-style and styles for papers and journal names and book titles. We tried an updated stylefile version without success and will therefore clarify with the editor or during final typesetting.*

References: is there no doi/URL for IUSS \ldots

*We checked again but couldn't find any.*

References: Meissner et al. is incomplete

References: Monteith is incomplete

References: Orchiston is incomplete

*Thank you very much for pointing out all the issues in the list of references. Where not indicated differently, the entry was revised and completed.*

*The editor is informed about my (friendly) long-term relationship with some of the co-authors.*

*PS: sorry for the LaTeX markups, I was not aware that HESS removed that option this year …*

**References**

*Beysens, D. (2018). Dew water. River Publishers.*

*Feigenwinter, C., Franceschi, J., Larsen, J. A., Spirig, R., & Vogt, R. (2020). On the performance of microlysimeters to measure non-rainfall water input in a hyper-arid environment with focus on fog contribution. Journal of Arid Environments, 182, 104260.*

*Florentin, A., & Agam, N. (2017). Estimating non-rainfall-water-inputs-derived latent heat flux with turbulence-based methods. Agricultural and Forest Meteorology, 247, 533-540.*

*Hannes, M., Wollschläger, U., Schrader, F., Durner, W., Gebler, S., Pütz, T., ... & Vogel, H. J. (2015). A comprehensive filtering scheme for high-resolution estimation of the water balance*

components from high-precision lysimeters. *Hydrology and Earth System Sciences, 19(8),* 3405-3418.

Kidron, G. J., & Kronenfeld, R. (2020b). Microlysimeters overestimate the amount of non-rainfall water–an experimental approach. *Catena, 194,* 104691.

Li, Y., Aemisegger, F., Riedl, A., Buchmann, N., & Eugster, W. (2021). The role of dew and radiation fog inputs in the local water cycling of a temperate grassland during dry spells in central Europe. *Hydrology and Earth System Sciences, 25(5),* 2617-2648.

Migliavacca, M., Perez‑Priego, O., Rossini, M., El‑Madany, T. S., Moreno, G., Van der Tol, C., ... & Reichstein, M. (2017). Plant functional traits and canopy structure control the relationship between photosynthetic $CO_2$ uptake and far‑red sun‑induced fluorescence in a Mediterranean grassland under different nutrient availability. *New Phytologist, 214(3),* 1078-1091.

Monteith, J. L. (1957). Dew. *Quarterly Journal of the Royal Meteorological Society, 83(357),* 322-341.

Qubaja, R., Amer, M., Tatarinov, F., Rotenberg, E., Preisler, Y., Sprintsin, M., & Yakir, D.: Partitioning evapotranspiration and its long-term evolution in a dry pine forest using measurement-based estimates of soil evaporation. *Agricultural and Forest Meteorology, 281,* 107831, 2020

Riedl, A., Li, Y., Eugster, J., Buchmann, N., & Eugster, W. (2022). High-accuracy weighing micro-lysimeter system for long-term measurements of non-rainfall water inputs to grasslands. *Hydrology and Earth System Sciences, 26(1),* 91-116.

Tomaszkiewicz, M., Abou Najm, M., Beysens, D., Alameddine, I., & El-Fadel, M. (2015). Dew as a sustainable non-conventional water resource: a critical review. *Environmental reviews, 23(4),* 425-442.

Verhoef, A., Diaz-Espejo, A., Knight, J. R., Villagarcía, L., and Fernández, J. E.: Adsorption of Water Vapor by Bare Soil in an Olive Grove in Southern Spain, *Journal of Hydrometeorology, 7,* 1011–1027, 2006

---

## Author Comment (AC3)

| hess-2021-519 | Submitted on Oct 2021 |
|---|---|
| Lysimeter based evaporation and condensation dynamics in a Mediterranean ecosystem | |
| Handling Editor | Lixin Wang |
| Manuscript type | Research article |
| Status | Final response (HESS discussion) |

**Author's response to Referee #1**

(Referee comments in black, *author's response in blue italic*)

**General comments**

This paper perfectly interprets the composition of non-rainfall water input using data from lysimeters and meteorological data. However, author pointed out that condensation processes are dew when water originates from the atmosphere or soil distillation when the water originates from the soil beneath. Therefore, difference between soil surface temperature and dewpoint temperature of nearby air is an important factor for dew. So, Dew was therefore assigned when $Ts < (Tdew - Tdew;t)$ where $Tdew;t$ is set to 1:4 â−¦C in Flux partitioning (Lines 162-175). Then, how to quantify the part of dew from soil. This part of water is only the process of water migration in soil and not the input of external water.

*We thank the referee for acknowledging the interpretation of our data and we are grateful for the comments posted that help us to clarify some assumptions that the analysis is based on.*

*There is a very well-written summary on the state of the art of quantification of soil distillation and the distinction from dew in the introduction of the manuscript from Li et al, 2021. In summary, they write that a combination of methods is always necessary for the quantification of soil distillation and its distinction from dew. Distillation water is either modeled by the vertical gradient method, relying on many assumptions (Monteith, 1957), or measured by using water stable isotopes in field campaigns. Li et al (2021) recommend combining lysimetric measurements with isotopic composition measurements to partition NRW from ambient water vapor and distillation.*

*Although one can argue indeed that the condensed water of soil origin shouldn't be counted as water input, the current literature lists it as NRW(I) (Li et al., 2021; Zhang et al., 2019a; Zhang et al., 2019b). Lysimeters readings, however, register the net water gain of the soil and plant monolith, and therefore mainly the input of external water (Meissner et al., 2007; Nolz et al., 2013). Because in theory, soil distillation shouldn't affect the lysimeter net weight if the water vapor condensing on the leaf surfaces stems from the soil below in the same lysimeter (Li et al., 2021). This is of course an assumption that might not always hold because it also depends on vapor transport characteristics and timing of the phase changes (evaporation from the capillaries, condensation on the leaves), but these speculations go beyond the scope of the analysis.*

*We fully agree with the reviewer that it is debatable whether soil distillation should be counted as water input or not. As we explained, however, in our setup the distinction is most likely of minor importance. Nevertheless, we understood from the reviewer's comment that this theoretical background should be clarified in the manuscript and we included this information in the Discussion to improve the understanding for the readers.*

**References**

Li, Y., Aemisegger, F., Riedl, A., Buchmann, N., & Eugster, W. (2021). The role of dew and radiation fog inputs in the local water cycling of a temperate grassland during dry spells in central Europe. Hydrology and Earth System Sciences, 25(5), 2617-2648.

Meissner, R., Seeger, J., Rupp, H., Seyfarth, M., & Borg, H. (2007). Measurement of dew, fog, and rime with a high‑precision gravitation lysimeter. Journal of Plant Nutrition and Soil Science, 170(3), 335-344.

Monteith, J. L. (1957). Dew. Quarterly Journal of the Royal Meteorological Society, 83(357), 322-341.

Nolz, R., Kammerer, G., & Cepuder, P. (2013). Interpretation of lysimeter weighing data affected by wind. Journal of Plant Nutrition and Soil Science, 176(2), 200-208.

Zhang, Q., Wang, S., Yue, P., & Wang, R. (2019a). A measurement, quantitative identification and estimation method (QINRW) of non-rainfall water component by lysimeter. MethodsX, 6, 2873-2881.

Zhang, Q., Wang, S., Yue, P., & Wang, S. (2019b). Variation characteristics of non-rainfall water and its contribution to crop water requirements in China's summer monsoon transition zone. Journal of Hydrology, 578, 124039.

---

## Author Response (AR2)

| hess-2021-519 | Submitted on Oct 2021 |
|---|---|
| Lysimeter based evaporation and condensation dynamics in a Mediterranean ecosystem | |
| Handling Editor | Lixin Wang |
| Manuscript type | Research article |
| Status | Final response (HESS discussion) |

**Author's response to anonymous Referee #1**

(Referee comments in black, *author's response in blue italic*)

**General comments**

The authors provide a complete description and photographs of the large lysimeter. The reader can better understand the observed process. As other reviewers have pointed out, the overestimation of NRW is due to the design of the instrument, not the technology itself. Although, instrumental and methodological uncertainty can affect results, it is more important to distinguish the composition of NRW and its occurrence time in arid areas, especially in different seasons. At present, more attention has been paid to NRW's role in water balance and single component in arid regions. But in the dry period without precipitation, its quantity and frequency are ignored. Disentangling individual NRW fluxes is an important way to find the controlling factors. This paper analyzes not only the composition of NRW and its controlling factors, but also the seasonal and diurnal contribution of NRW to water input. The study is of great significance to the survival of organisms in arid areas. So, the paper is acceptabl.

*We thank the reviewer for taking the time and for the appreciation of our work.*

| hess-2021-519 | Submitted on Oct 2021 |
|---|---|
| Lysimeter based evaporation and condensation dynamics in a Mediterranean ecosystem | |
| Handling Editor | Lixin Wang |
| Manuscript type | Research article |
| Status | Final response (HESS discussion) |

**Author's response to Referee #2 Giora Kidron Report #2**

(Referee comments in black, *author's response in blue italic*)

**General comments**

The authors presented an interesting, well written and a thorough account presenting data from a set of lysimeters in central Spain. The data presented are important and I enjoyed reading the ms.

*We thank the reviewer for the appreciation of our work.*

Saying that, and especially due to the potential importance of the ms, there are still some issues that require clarification.

**Major points**

1. A major possible drawback may stem from the inherent failure of the lysimeter to adequately mimic the heat regime of the intact soil. This issue was recently addressed in Agric Forest Met (2021). While large lysimeters may have an advantage over microlysimeters in this respect, I wonder if the authors compared the temperatures within the lysimeter and the intact soil, particularly the surface temperatures. Whether undertaken or not, this information should be provided in the Discussion.

*We thank the reviewer for bringing up this potential issue, and we agree that this should be elaborated on in the manuscript. We prefer to avoid an additional comparison between the temperature inside and outside of the lysimeters, particularly because the respective surface temperature measurements are not colocated and hence not directly comparable. Therefore we preferred to include a paragraph in the discussion ($T_s$ denotes surface temperature) (lines 358 - 363):*

*"Despite the current version of large weighing lysimeters being designed with lower boundary control in order to prevent temperature deviations from the surrounding soil, the extent to which heat loss through their walls will affect NRW has yet to be evaluated at sites with suitable instrumentation, particularly since NRW is sensitive to deviations in $T_s$ (Yokoyama et al., 2021). Such a comparison was not possible in our instrumental setup because $T_s$ was derived from measurements in one location only, which was situated outside the lysimeters (Figure A1)."*

And in this regard, it is not clear what Ts stands for and how it was measured.

*$T_s$ stands for surface temperature and is already introduced in line 25 since it is the first time mentioned in that section (introduction). Its calculation, however, is described in line 115 with the referring equation that is given in Appendix A2.*

Furthermore, did the authors calculate the Td in accordance with the comparison done between the temperature sensors at 1 and 0.1 m? If this is the case, the dew data reflect possible condensation at 10 cm height above ground, which may adequately represent short grassland. However, it does not adequately represent the soil surface which will exhibit warmer temperatures. This issue should be discussed in the Discussion section.

*Thank you very much for your question and suggestion. Indeed we calculate $T_{dew}$ in accordance with the comparison, as described in line 169. Motivated by the reviewers suggestion we added the following sentence to the Discussion section (lines 92-394)*

*"The chosen temperature offset added to $T_{dew}$ was used in this study to account for the mean vegetation height. The soil surface, however, will exhibit warmer temperatures during the day. The height of the condensation surface should therefore always be considered."*

2. Some of the figures provide a general picture, but more precise data will be of great help. Please provide detailed composite graphs similar to Figure 3, in which rain, fog, dew and vapor are indicated. Please provide one graph (graph A) with data during a week without rain and graph B with data that include a rain event during the beginning of the week and data that shows the distribution of the different types of NRW during the remaining week. These graphs will help the reader to assess the differences between dewfall and distillation.

*As suggested by the reviewer we included the following sentence and the respective figure in the Appendix:*

*"Example lysimeter weight evolution and the assigned flux categories for a week with and without rain is shown in Figure D1 in the Appendix."*

[Figure]

*Figure D1. Lysimeter weights (normalised over the shown time period for each lysimeter by subtracting the respective minimum weight) exemplary shown for (a) a week without rain from August 15th. to August 20th., 2019, and (b) a week with several rain events from October 18th., to October 23th., 2019. The color code shows the respective flux that was assigned to the preceding weight change. Grey shaded areas illustrate nights.*

3. Please indicate in table (or graph) the monthly distribution of the number of days in which the daily NRW was >0.1 mm. 0.1-0.2 mm, >0.2 mm; indicate also the maximum recorded daily NRW. Was it linked to rain?

*We included the graph based on your suggestion with the following description of its content:*

[Figure]

*Figure E1. The number of days per month with average NRW input < 0.1, 0.1 - 0.2, and > 0.2 mm d⁻¹.*

*"On most days, the average NRWI is < 0.2 mm/day (Figure E1). Yields of > 0.2 mm per night are observed on 54 days distributed throughout the year. In late summer NRWI is predominantly between 0.1 and 0.2 mm day¹ while over the winter, the yields are typically*

*small and < 0.1 mm day$^{-1}$. The maximum recorded daily NRWI was 2.42 mm day$^{-1}$ on May 13th, 2020 between several small, consecutive rain events."*

4.     Discussion. Please assess the possible use of NRW by plants, biocrusts and soil microorganisms.  Also, expand on the relationships between lysimeter measurements and calculations carried out by eddy covariance.

*Based on the suggestion of the reviewer we expanded the paragraph (lines 372 - 378) to have a more detailed assessment of the possible use of NRW. It is now moved to lines 412-434 in order to avoid splitting the discussion about the decision tree-based NRW classification. Since biocrusts are not present in Majadas, which is now reported in line 85, we don't expand on this life form.*

*"The role of dew has often been reported as moistening plant surfaces with direct leaf water uptake (Tomaszkiewicz et al., 2015). Our results show that in Majadas, dew occurs predominantly at a time of the year when top SWC ranges between relatively wet values of 20 to 35 %. Therefore, we assume that in this ecosystem dew as plant water supply is generally less relevant as has been reported for desert vegetation (e.g. Hill et al., 2015). Dew may benefit plants indirectly for example by cooling the leaves during early summer or facilitating nutrient uptake over leaf tissues (Dawson and Goldsmith, 2018). Although we only investigated grassland, this could also be relevant for Quercus ilex, which is confirmed to allow water penetration from the upper leaf surface into the leaf interior (Fernández et al., 2014).*

*As opposed to dew, soil vapor adsorption occurs at low Ψ when grassland in Majadas has already senesced. Therefore, a potential ecological relevance would rather be to enhance microbial activity and trigger respiration from soil or (standing) litter (Evans et al., 2019; Gliksman et al., 2017; Dirks et al., 2010, McHugh et al., 2015). However, the highest $CO_2$ emissions during summer in terms of volume are expected to be caused by rain pulses (López-Ballesteros et al., 2016). Reports exist also on nighttime $CO_2$ uptake during adsorption but the underlying biogeochemical processes are not yet clear (Lopez-Canfin et al., 2022). Future research is necessary to disentangle and quantify the ecosystem response to the different types of NRW."*

**Minor points**

1.     L 35-37. Please expand. Condensation at RH <100%?

*We thank the reviewer for pointing out that this sentence is not clear. We expanded the sentence and hope it is more precise now (lines 34 - 40):*

*"In close contact with soil, the water vapor in the soil air is influenced by capillary and adsorptive forces (Tuller et al., 1999). These forces increase in relevance as the soil dries out. They reduce the saturation vapor pressure ($e_{sat}$, kPa) as a function of soil dryness (Edlefsen et al., 1943), leading to an earlier phase change of water from vapor to liquid within the soil, compared to free air. Under such conditions, the rH value of the near-surface air outside the soil  is well below 100 %, and condensation processes are classically not considered, but soil air can condensate."*

2.	L. 80. What is the source of vapor for dewfall?

*We are unsure how this information is relevant in line 80 (Section Site description), maybe the reviewer meant to refer to a different line? Furthermore, to our knowledge, this question cannot be answered without a field campaign using water stable isotopes which is beyond the scope of this study.*

3.	L 85. Please describe the growth dynamics of the vegetation along the year, specifically cover and height above ground. Also indicate whether biocrusts are present in the site.

*We thank the referee for this suggestion. We tried to include it as specifically as possible. However, since we don't have direct measurements of the fractional cover of our herbaceous vegetation, we included the information on fractional vegetation cover indirectly by reporting the leaf area index (lines 83-93).*

*"The herbaceous layer consists of native annual grasses, forbs, and legumes (Migliavacca et al., 2017). The growing season for the herbaceous layer begins after the first rains after summer (typically in mid-October) and is inhibited by low temperatures in winter before peaking in spring before the dry season (Luo et al., 2020). During the dry season, the herbaceous species are inactive until the return of rain (Perez-Priego et al., 2018), and bare soil is visible below the dry biomass. The mean leaf area index of the herbaceous vegetation ranges between $0.25 \pm 0.07$ $m^2$ $m^{-2}$ in summer to $1.75 \pm 0.25$ $m^2$ $m^{-2}$ at the peak of the growing season in spring (El-Madany et al., 2021) with the same seasonal dynamics in vegetation height that varies between 0.05 m and 0.20 m (Migliavacca et al. 2017). Biocrusts are not present at the site."*

4.	L 100. "soil temperature is controlled with a heat exchange system". Please expand.

*We have added the following sentence (lines 110-112):*

*"Buried at the bottom of each lysimeter, 6 m tubing systems are connected to tubing systems buried at the same depth in the surrounding soil. Pumping of water through the system in a closed loop regulates the soil temperature within each lysimeter column by heat exchange (Podlasly & Schwärzel, 2013)."*

5.	l 117. Define Ts

*This issue has been already addressed above in our reply to major point 1. Thank you.*

6.	Fig. 3: please check the compatibility between 3a and 3d. For instance, while 3a showes very little rain between June and August, Fig. 3d points at a high contribution of rain.

*Indeed the amount of rain in 3a is small in comparison to the contribution of rain in some winter weeks, however, in relation to the amount of adsorbed water from the soil it is still much larger.*

---

## Author Response (AR3)

| hess-2021-519 | Submitted on Oct 2021 |
|---|---|
| Lysimeter based evaporation and condensation dynamics in a Mediterranean ecosystem | |
| Handling Editor | Lixin Wang |
| Manuscript type | Research article |
| Status | Final response (HESS discussion) |

**Author's response to Referee #2 Giora Kidron Report #3**

(Referee comments in black, *author's response in blue italic*)

**General comments**

The authors presented a thorough, well written account presenting data from a set of lysimeters in central Spain. I warmly recommend publication in HESS pending minor revision.

*We kindly thank the reviewer for acknowledging our work and for the additional suggestion that helped to improve clarity.*

**Main points**

a.      Figure 3d clearly shows a clear disparity between the amount of dew and fog during the dry and the wet seasons. The authors should add a table that shows the data in the dry and the wet season. The table should include the number and the average amount of rain, dew, fog and vapor adsorption events for each season.
The discussion should also relate to these differences and the possible effect of the vegetation cover and height on the data.

*On a related point in the previous revision of the manuscript, we have added Figure E1 such that we feel an additional table would duplicate information.*

*As the reviewer points out, the disparity between the amount of dew and fog during the dry and the wet season is clearly shown in Figure 3, and this seasonal disparity is additionally depicted in Figure 2.*

*In order to accommodate this comment, we have added a clarification about the observed water flux differences between the dry and wet seasons (lines 264 - 267 and  275-280 ) and the respective role of changing vegetation cover (lines 383 - 391).*

*Now the revised manuscript reads:*

*Lines 265 - 268:*

*"On most days, the average NRWI is < 0.2 mm $d^{-1}$ (Fig. E1). Yields of < 0.2 mm $d^{-1}$ are observed on 54 days distributed throughout the year. During the dry season NRWI is predominantly between 0.1 and 0.2 mm $d^{-1}$ while over the wet season, the yields are typically small and < 0.2 mm $d^{-1}$."*

*Lines 275-280:*

*"The weekly sums in Fig. 3 illustrate that the seasonal dynamics are consistent across lysimeters. The ecosystem receives atmospheric water at any time of the year but with shifting relative relevance of the water flux types during the wet and the dry season.*

*Rain is the dominant liquid water input (i.e. it contributes more than 50 % of the weekly water input) during 29 weeks since its total amount is usually much greater than NRW inputs, whereas NRW inputs are dominant in 24 weeks. They are even the only water input during 15 weeks with adsorption as the exclusive water input in 10 weeks of the year during the dry season.*

*During the dry season, the median contribution of adsorption is 0.9 mm $week^{-1}$ whereas dew and fog contribute < 0.5 mm $week^{-1}$. With ET amounting on average to 5.7 mm $week^{-1}$, thus adsorption compensates on average for 19 % of the weekly water loss through ET, ranging from 8.0 % at the beginning of June to 42.5 % at the beginning of September. The median contribution of dew and fog is 0.43 and 0.38 mm $week^{-1}$, respectively. Dew and fog occurrence is synchronized with rain with regard to the seasonal occurrence (in the wet season), and therefore their relative contribution to the water input on a weekly scale is small (see Fig. 3d)."*

*Lines 383 - 391:*

*"Vegetation height and density affect the radiative exchange of the soil surface and the plant canopy affecting dew formation occurrence and amount (Monteith and Unsworth, 2013). Xiao et al. (2009), for example, observed a positive relationship between maize plant height and the amount of dew formed per night over two years. The grass height and LAI in Majadas change over the season reaching a maximum in height and density usually in winter and spring (Migliavacca et al.; 2017). This is in line with our findings of the occurrence of dew and fog nearly exclusively during the wet season (although it is not possible to separate the effects from meteorological conditions). Unfortunately, no data on plant height and density were collected continuously in the lysimeters columns, which would have allowed studying their effect on the amount of dew formation (and fog deposition). However, due to the active vegetation on the lysimeters, the effect of their variation in height and density is represented in our results."*

b.       The discussion is long and fluctuates between sections that describe the daily and seasonal distribution, the ecological importance of the findings and technical issues. To increase clarity, I suggest the addition of subtitles to the Discussion in accordance to the main themes that are discussed.

*We agree with the reviewer and gratefully followed her suggestion. We restructured and divided the Discussion into four subsections with the following titles:*

   A.  *4.1 Non-rainfall water frequency, duration, and amounts*
   B.  *4.2 Impact of methodological uncertainty*
   C.  *4.3 Distinguishing non-rainfall water formation*
   D.  *4.4 Ecological relevance and open questions*

*Some parts within the discussion needed rephrasing to match their new position in the text but the content of the Discussion was not changed apart from the sentences cited above, under Major points a).*

**Minor points**

a.       l. 80. Add potential evaporation.

*We followed the suggestion of the reviewer and added potential evaporation (l. 80):*

*"The mean annual potential evapotranspiration calculated with the Priestley-Taylor equation (Knauer et al.; 2018) amounts to 1152 mm  year$^{-1}$."*

b.       In keeping with publications highlighting the importance of the construction material, please indicate the construction material of the lysimeter.

*We followed the suggestion of the reviewer and added the material of the lysimeter (l. 99):*

*"The site is equipped with three lysimeter stations, each containing two weighable high-precision polyethylen-high-density lysimeters (UGT, Müncheberg, Germany), for a total of 6 columns."*

---

## Author Response (AR4)

| hess-2021-519 | Submitted on Oct 2021 |
|---|---|
| Lysimeter based evaporation and condensation dynamics in a Mediterranean ecosystem | |
| Handling Editor | Lixin Wang |
| Manuscript type | Research article |
| Status | Final response (HESS discussion) |

**Author's response to remarks from the preceding review file validation**

(Referee comments in black, *author's response in blue italic*)

1. Regarding your figure E1: With the next revision, please check if a copyright statement/image credit is required and add it to the figure caption, if applicable. If you are the originator, you can just inform us.

*For Figure E1, no copyright statement is required. I am the originator of this figure*

2. Please ensure that the colour schemes used in your maps and charts allow readers with colour vision deficiencies to correctly interpret your findings. Please check your figures using the Coblis – Color Blindness Simulator (https://www.color-blindness.com/coblis-color-blindness-simulator/) and revise the colour schemes accordingly.

*All Figures have been checked with the suggested Coblis - Color Blindness Simulator. We changed the color assigned to one flux (adsorption) in Fig. 2; Fig. 3; and Fig. 7 accordingly to ease the distinction between the fluxes.*